# Flood spatial coherence, triggers and performance in hydrological simulations: large-sample evaluation of four streamflow-calibrated models.

Manuela I. Brunner[1], Lieke A. Melsen[2], Andrew W. Wood[1,3], Oldrich Rakovec[4,5], Naoki Mizukami[1], Wouter J. M. Knoben[6], and Martyn P. Clark[6]

[1]Research Applications Laboratory, National Center for Atmospheric Research, Boulder CO, USA
[2]Hydrology and Quantitative Water Management, Wageningen University, Wageningen, Netherlands
[3]Climate and Global Dynamics Laboratory, National Center for Atmospheric Research, Boulder CO, USA
[4]Department Computational Hydrosystems, Helmholtz Centre for Environmental Research, Leipzig, Germany
[5]Faculty of Environmental Sciences, Czech University of Life Sciences Prague, Praha – Suchdol, Czech Republic
[6]University of Saskatchewan Coldwater Laboratory, Canmore, Canada

**Correspondence:** Manuela I. Brunner (manuelab@ucar.edu)

**Abstract.** Floods cause large damages, especially if they affect large regions. Assessments of current, local and regional flood hazards and their future changes often involve the use of hydrologic models. A reliable hydrologic model ideally reproduces both local flood characteristics and spatial aspects of flooding under current and future climate conditions. However, uncertainties in simulated floods can be considerable and yield unreliable hazard and climate change impact assessments. This study evaluates the extent to which models calibrated according to standard model calibration metrics such as the widely-used Kling–Gupta efficiency are able to capture flood spatial coherence and triggering mechanisms. To highlight challenges related to flood simulations, we investigate how flood timing, magnitude and spatial variability are represented by an ensemble of hydrological models when calibrated on streamflow using the Kling–Gupta efficiency metric, an increasingly common metric of hydrologic model performance also in flood-related studies. Specifically, we compare how four well-known models (SAC, HBV, VIC, and mHM) represent (1) flood characteristics and their spatial patterns; and (2) how they translate changes in meteorologic variables that trigger floods into changes in flood magnitudes. Our results show that both the modeling of local and spatial flood characteristics is challenging as models underestimate flood magnitude and flood timing is not necessarily well captured. They further show that changes in precipitation and temperature are not necessarily well translated to changes in flood flow, which makes local and regional flood hazard assessments even more difficult for future conditions. From a large sample of catchments and with multiple models, we conclude that calibration on the integrated Kling–Gupta metric alone is likely to yield models that have limited reliability in flood hazard assessments, undermining their utility for regional and future change assessments. We underscore that such assessments can be improved by developing flood-focused, multi-objective and spatial calibration metrics, by improving flood generating process representation through model structure comparisons, and by considering uncertainty in precipitation input.

# 1  Introduction

Many studies use a hydrological model driven by present or future meteorological forcing data to derive flood estimates for current and future conditions. However, data, model structure, and parameter uncertainties can be considerable (Clark et al., 2016) especially when considering extreme events such as floods (Brunner et al., 2019b; Das and Umamahesh, 2018) and when considering hydrological change. It is therefore challenging to produce statistically reliable estimates of future changes in flood hazard.

A model ideally reproduces different aspects of flooding, including local characteristics such as event magnitude and timing. To obtain such satisfactory flood simulations, hydrological models are often calibrated using one or several objective functions. One widely-used metric that is often used in flood studies (e.g. Hundecha and Merz, 2012; Köplin et al., 2014; Vormoor et al., 2015; Wobus et al., 2017) is the Nash–Sutcliffe efficiency ($E_{NS}$; Nash and Sutcliffe 1970) because it is considered integrative compared to others and focuses attention on high flows. However, $E_{NS}$ is formulated so that its optimal value systematically underestimates flow variability (Gupta et al., 2009), undermining the ability of a model to reproduce peak flow values. A related metric, the Kling–Gupta efficiency ($E_{KG}$; Gupta et al. 2009), is free from this constraint and may improve simulations of peak flows, especially if the variability related component of the score is emphasized in calibration (Mizukami et al., 2019). This metric has been frequently used in recent flood modeling studies (e.g. Harrigan et al., 2020; Hirpa et al., 2018; Huang et al., 2018; Thober et al., 2018; Brunner and Sikorska, 2018) and seems to be widely accepted as a suitable choice for flood studies. This may arise from the general practice of developing models for a range of objectives. However, recent studies have shown that capturing flood magnitude and timing is challenging when such standard calibration metrics are used for parameter estimation (Lane et al., 2019; Brunner and Sikorska, 2018; Mizukami et al., 2019).

In addition to simulating the timing and magnitude of flow at individual catchments, it is also important to realistically reproduce spatial dependencies, i.e. the relationship of flood occurrence across gauging stations (Keef et al., 2013; De Luca et al., 2017; Berghuijs et al., 2019). An over- or underestimation of spatial dependencies across a network of gauging stations in regional flood hazard and risk assessments has been shown to under- or overestimate regional damage, respectively (Lamb et al., 2010; Metin et al., 2020). Prudhomme et al. (2011) have shown for a set of large-scale hydrological models that simulated high flow episodes are less spatially coherent than observed events. Despite their high relevance for impact, the spatial aspects of flooding have often been overlooked in past simulation studies.

Local and spatial flood characteristics should be reliably simulated not only under current but also under future climate conditions. However, models calibrated for current conditions may not be transferable in time (Thirel et al., 2015) partly because of a sub-optimal representation of flood producing mechanisms. To overcome this transferability problem, the differential split-sample test has been proposed, where the model is calibrated and validated on two periods with differing climate conditions (Klemes, 1986; Seibert, 2003).

In this study, we evaluate the extent to which model calibrated according to the widely-used model calibration metric $E_{KG}$ are able to capture flood spatial coherence and flood triggering mechanisms. To this end, we first evaluate how well different hydrological models capture local flood events following the current paradigm, secondly we expand the evaluation by analyz-

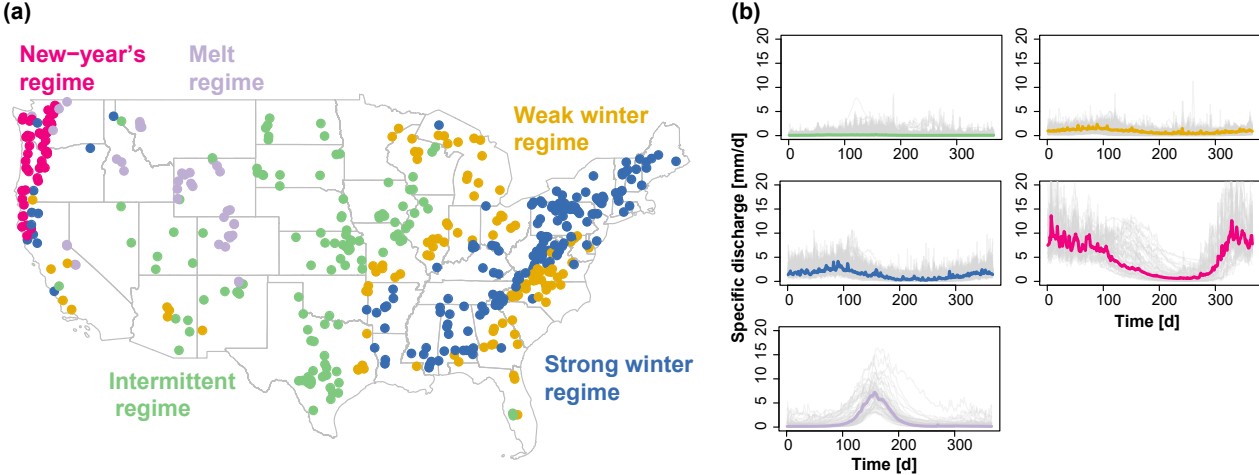

**Figure 1.** a) Map of the 488 catchments in the conterminous United States belonging to the five regime classes indicated by their gauge location: 1) Intermittent, 2) weak winter, 3) strong winter, 4) New Year's, and 5) melt. b) Median regime per regime class (colored lines) and variability of regimes within a class (one line per catchment, grey) (Brunner et al., 2020b).

ing how well the models capture spatial flood dependence, and finally we evaluate how the models capture flood triggering mechanisms. With this thorough evaluation, we assess which aspects of hydrological models may need to be improved if we want to bring hazard and change impact assessments to a point where we can make more reliable assessments of regional flood hazard and future changes.

For documenting modeling challenges related to floods, we look at the model output of four widely used hydrological
models (Addor and Melsen, 2019), namely, the Sacramento Soil Moisture Accounting model (SAC-SMA; Burnash et al., 1973) combined with SNOW–17 (Anderson, 1973), the Hydrologiska Byråns Vattenbalansavdelning model (HBV; Bergström, 1976), the Variable Infiltration Capacity model (VIC; Liang et al., 1994), and the mesoscale hydrologic model (mHM; Kumar et al., 2013; Samaniego et al., 2010). Identifying and documenting model weaknesses regarding regional and future flooding will highlight avenues for future model development and reveal potential deficiencies of a calibration strategy often applied for
research studies on floods.

## 2   Data and Methods

To study how local and spatial flood characteristics are reproduced by hydrological models calibrated on streamflow using the individual calibration metric, $E_{KG}$, we compare observed to simulated flood event characteristics for a set of 488 catchments in the conterminous United States that have minimal human impact and catchment areas ranging from 4 to 2000 km$^2$ (Figure
1a) (Newman et al., 2015b). The dataset comprises catchments with a wide range of climate and streamflow characteristics

ranging from catchments with intermittent regimes and a very weak seasonality to catchments with a very strong seasonal cycle under the influence of snow (New Year's and melt regimes; Figure 1b; Brunner et al. 2020b). Observed streamflow time series are available from the U.S. Geological Survey (USGS, 2019).

## 2.1 Model simulations

We use daily streamflow simulations for the period 1981–2008 generated with four well-known hydrological models (Addor and Melsen, 2019) offering different model structures and complexity: the lumped SAC model (Figure A1; Burnash et al., 1973), the lumped HBV model (Figure A2; Bergström, 1976), the lumped version of the VIC model (Figure A3; Liang et al., 1994), and the grid-based, distributed mesoscale hydrologic model mHM (Figure A4; Kumar et al., 2013; Samaniego et al., 2010). The model parameters were calibrated on streamflow observations by minimizing $E_{KG}$ by Melsen et al. (2018) using

Sobol-based Latin hypercube sampling (Bratley and Fox, 1988) for SAC, HBV, and VIC and by Mizukami et al. (2019) for mHM using multi-scale parameter regionalization where the transfer function parameters were identified using the dynamically dimensioned search algorithm (Tolson and Shoemaker, 2007). $E_{KG}$ is defined as:

$$E_{KG}(Q) = 1 - \sqrt{[s_\rho \cdot (\rho - 1)]^2 + [s_\alpha \cdot (\alpha - 1)]^2 + [s_\beta \cdot (\beta - 1)]^2}, \tag{1}$$

where $\rho$ is the correlation between observed and simulated runoff, $\alpha$ is the standard deviation of the simulated runoff divided

by the standard deviation of observed runoff, and $\beta$ is the mean of the simulated runoff, divided by the mean of the observed runoff. $s_\rho$, $s_\alpha$, and $s_\beta$ are scaling parameters enabling a weighting of different components. When used individually, $E_{KG}$ has been found to result in a better performance for annual peak flow simulation than the long-standing and related hydrologic model evaluation metric $E_{NS}$ (Mizukami et al., 2019).

For SAC, Melsen et al. (2018) calibrated and evaluated 18 out of the 35 parameters available in the coupled Snow-17

and SAC-SMA modeling system, for HBV 15 parameters, for VIC 17 parameters, and for mHM Rakovec et al. (2019) and Mizukami et al. (2019) calibrated and evaluated up to 48 parameters. All the models were driven with daily, spatially lumped meteorological forcing data representing current climate conditions: SAC, HBV, and VIC were driven with Daymet meteorological forcing (1 km resolution; Thornton et al., 2012) and mHM with the forcing by Maurer et al. (2002) (12 km resolution) both derived from observed precipitation and temperature. SAC, HBV, and VIC were calibrated and evaluated on the period

1985–2008 while mHM was calibrated on the period 1999–2008 and evaluated on the period 1989–1999. After calibration, all four models were run for the period 1980–2008 (calendar years), where the period 1980–1981 was here used for spin-up and therefore discarded from the analysis.

To provide insights with respect to where model performance is better/worse, we provide model evaluation results for five different streamflow regime types, which have been shown to be distinct in their flood behavior: 1) Intermittent, 2) weak winter,

3) strong winter, 4) New Year's, and 5) melt (Figure 1; Brunner et al., 2020b). Catchments with intermittent regimes experience floods mainly in spring and summer, those with weak winter regimes in winter and spring, those with strong winter regimes

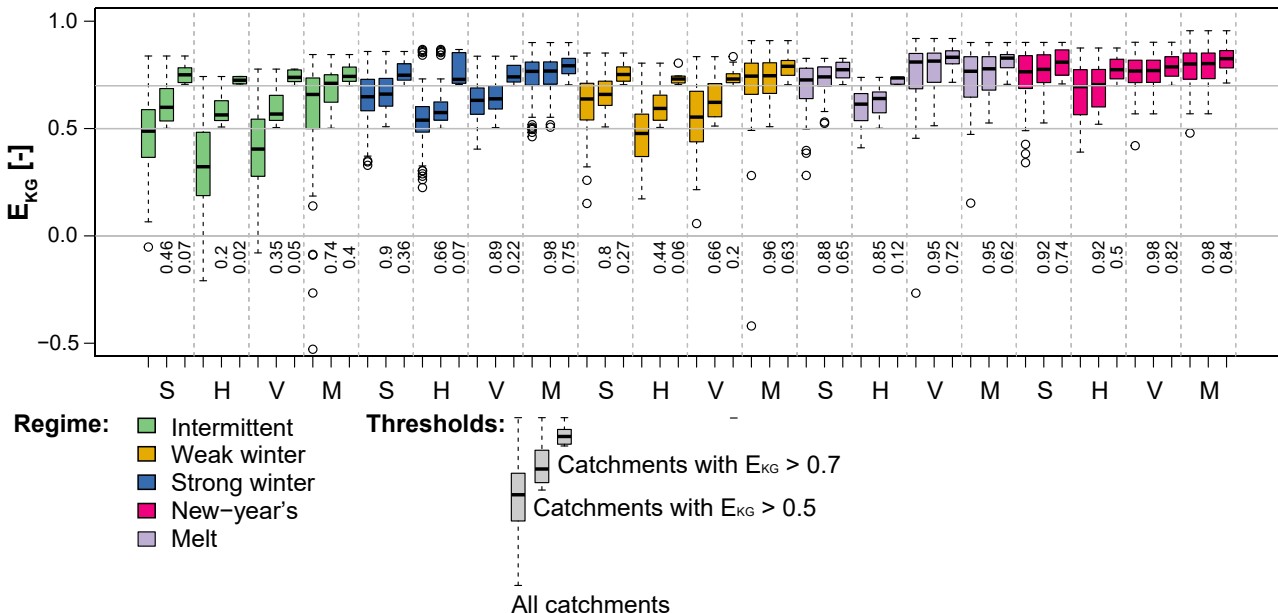

**Figure 2.** Model performance in terms of $E_{KG}$ over the period 1981–2008 for the four models SAC (S), HBV (H), VIC (V), and mHM (M) per hydrological regime: intermittent (114 catchments), weak winter (108), strong winter (176), New Year's (50), and melt (40). For each model and regime, three boxplots are shown: all catchments, catchments with $E_{KG} > 0.5$, and catchments with $E_{KG} > 0.7$. The percentage [-] of catchments of a regime class above the corresponding threshold is indicated below the 0 line.

in winter, those with a New Year's regime around New Year, and those with a melt-dominated regime in spring because of snowmelt.

Model performance in terms of $E_{KG}$ varies spatially and is related to the hydrological regime (Figure 2). It is overall lowest

for catchments with intermittent regimes and a weak seasonality and highest for catchments with a strong seasonality such as a melt and New Year's regime. However, there is a high within-class variability in model performance. The finding that intermittent regimes are challenging to model successfully is well known in hydrology and reproduced in many studies, e.g., Unduche et al. (2018), who show that hydrological modeling on Prairie watersheds is very complex (Hay et al., 2018). Intermittent regimes may suffer in calibration if they rely solely on correlation-type measures because their day to day variation is

more difficult to reproduce than a more pronounced and regular seasonality. Overall model performance decreases from mHM (median $E_{KG}$ 0.69), over SAC (median $E_{KG}$ 0.63) and VIC (median $E_{KG}$ 0.60) to HBV (median $E_{KG}$ 0.52). In addition to streamflow, we use areal precipitation and simulated soil moisture to explain potential differences in model performance.

### 2.2 Model evaluation for floods

We compare local and spatial flood characteristics extracted from the observed time series to those of the series simulated

with the four models for the period 1981–2008 for the five streamflow regimes introduced above. Such a comparison enables

identification of flood characteristics whose model representation could potentially be improved. To better understand potential model deficiencies, we look at how models capture flood triggering mechanisms and how they simulate floods under climate conditions different from the current ones.

### 2.2.1 Flood event identification

Flood events are identified for each of the five time series (one observed, four simulated) using a peak-over-threshold (POT) approach similar to the one used in Brunner et al. (2019a, 2020b). This approach consists of two main steps and results in two data sets each, which are used for the local and spatial analysis, respectively: (1) POT events (i.e. peak discharges) in individual catchments and (2) event occurrences across all catchments. In Step 1, independent POT events are identified in the daily discharge time series of the individual catchments using the 25[th] percentile of the corresponding time series of annual maxima as a threshold (Schlef et al., 2019) and by prescribing a minimum time lag of 10 days between events (Diederen et al., 2019). This procedure results in a first quartile of 36, a median of 40, and a third quartile of 47 events identified per basin. In Step 2, a data set consisting of the dates of flood occurrences across all catchments is compiled. This set is converted into a binary matrix which specifies for each catchment (columns) whether or not it is affected by a specific event (rows). We consider a catchment to be affected by a certain event if it experiences an event within a window of $\pm 2$ days of that event to take into account travel times. In addition to a binary matrix of all events, we set up seasonal binary matrices (winter: Dec–Feb, spring: Mar-May, summer: June–Aug, fall: Sept–Nov).

### 2.2.2 Flood characteristics at individual sites

We use the data sets resulting from Step 1, the POT events at individual catchments, to evaluate how well the models reproduce flood statistics at individual sites. We focus on the total number of events $n$ (actual error: $n_s - n_o$, where $s$ represents simulations and $o$ observations), magnitude in terms of mean peak discharge $x$ (relative error: $(x_s - x_o)/x_o$), and mean timing (absolute error: circular statistics suitable for defining central tendencies of variables with a cycle (Burn, 1997)).

### 2.2.3 Spatial flood dependence

We then use the data sets resulting from Step 2 to evaluate how models reproduce overall and seasonal spatial flood dependence. To do so, we use the connectedness measure introduced by Brunner et al. (2020a), which quantifies the number of catchments with which a specific catchment co-experiences floods. The number of concurrent flood events for a pair of stations is determined based on a data set consisting of the dates of flood occurrences across all catchments. This set is converted into a binary matrix which specifies for each catchment whether or not it is affected by a certain event. The matrix compiled using observed streamflow time series contained 1164 events among which 258 occur in winter, 291 in spring, 324 in summer, and 291 in fall. Following the definition used by Brunner et al. (2020a), a catchment is connected to another catchment if they share a certain number of events. We here used an event threshold of 1% of the total or seasonal number of events to define

connectedness (all months: 12 events, seasons: 3 events). We computed actual errors in flood connectedness by subtracting observed from simulated connectedness over all seasons and per season.

### 2.2.4 Flood triggers

To explain potential differences in model performance, we look at the relationship of simulated peak discharge with the two flood triggers precipitation and soil moisture on the day of flood occurrence. We focus on the day of occurrence because time of concentration is typically small for small headwater basins (USDA-NRCS, 2010).

### 2.2.5 Floods under change

In addition to assessing model performance under current climate conditions, we would like to understand potential, additional challenges arising when interested in future conditions. To do so, we look at how models translate changes in event temperature and precipitation into changes in POT discharge by performing a resampling-based sensitivity analysis. This sensitivity analysis aims at evaluating whether a model is still reliable under climate conditions different from the ones used in model calibration similar to split-sample or differential split-sample calibration/validation schemes (Klemes, 1986; Coron et al., 2012; Refsgaard et al., 2014; Thirel et al., 2015). To perform this sensitivity analysis, we generate surrogate time series of temperature, precipitation, and streamflow for each catchment (Wood et al., 2004; Brunner et al., 2020b). To generate these series, we randomly sample a series of years with replacement in the period 1981–2008 which we use to compose time series consisting of the daily values corresponding to these years for each of the three variables. For each of the surrogate series, we again extract POT flood events using the same procedure as described under Step 1. For each of the extracted events we then determine temperature and precipitation on the day of peak discharge. We use the sets of peak discharge, event temperature and event precipitation to compute mean event discharge, temperature, and precipitation, which enables the derivation of a relationship between mean POT discharge and the two meteorological variables during events. We repeat the resampling $n = 500$ times to derive a relationship between changes in mean event temperature and precipitation and changes in mean POT streamflow. This resampling experiment results in a response surface of POT discharge spanned by mean event temperature and mean event precipitation for each catchment. We summarize the results obtained at individual locations by computing horizontal and vertical sensitivity gradients on these reaction surfaces using a linear regression model. The horizontal gradient describes the strength of POT discharge changes in response to event temperature changes while the vertical gradient describes the strength of change in response to changes in event precipitation. Conducting this experiment for both observed and simulated time series allows for the determination of whether the models react to changes in mean event temperature and precipitation in the same way as the real world system and are therefore suitable for the use in climate change impact assessments on floods. If models produce different climate sensitivities than the ones seen in the observations, the use of models to simulate sets of flood events for future conditions may preclude reliable change assessments.

## 3 Results

### 3.1 Flood characteristics at individual sites

Model performance at individual sites with respect to the number of events, event magnitude, and timing varies by model and hydrological regime type (Figure 3). For most catchments, the median deviation between the simulated and observed number of flood events lies close to zero (SAC: -3 events, HBV: -1, VIC: -1, mHM: 0). However, the simulations result in over- and underestimations of the number of events depending on the catchment (1st and 3rd quartiles for SAC: -9, 4; HBV: -8, 15; VIC: -7, 6; mHM: -6, 6). The overestimation is strongest for HBV, which overestimates the number of events for catchments with intermittent, weak winter, and melt regimes (Brunner et al., 2020b). Event magnitude in terms of peak discharge is generally underestimated for all regime types independent of the model and also absolute flood timing errors are present in all models. They are the highest in catchments with intermittent regimes with a high variability in flood timing and low in catchments with a New Year's and melt regime where the flood season is limited to a few months (Brunner et al., 2020a).

### 3.2 Spatial flood dependencies

Over all seasons, most models show a median error close to zero for flood connectedness. Flood connectedness can be over- and underestimated dependent on the catchment by most of the models while HBV overestimates spatial dependence in most catchments (Figure 4). Seasonally, most models over- or underestimate spatial dependence in certain regions. In winter, connectedness is overestimated by most models except for VIC and the strength of overestimation is strongest for HBV. In spring, most models tend to underestimate spatial dependence except for HBV that results in an overestimation of spatial dependence for catchments with an intermittent regime. Connectedness overestimation by HBV is most pronounced for catchments with an intermittent regime. Otherwise, connectedness over-/underestimation seems to be independent of the regime.

### 3.3 Flood triggers

The differences in model performance regarding local and spatial flood characteristics may be partially explained by differences in their structure and how they transform precipitation into runoff. Figure 5 shows how simulated peak discharge is related to event precipitation, event precipitation plus snowmelt, and simulated soil moisture over all catchments for the four hydrologic models. The SAC and VIC models show similar simulated relationships for all three variable pairs. There is a positive relationship between peak discharge and precipitation and peak discharge and rainfall plus snowmelt, i.e. the higher the precipitation input or rainfall and snowmelt combined, respectively, the higher the resulting peak discharge. This relationship is slightly more expressed for VIC than for SAC. In both models, soil moisture and event magnitude are also positively related with lower peak values potentially associated with lower soil moisture states than more severe events. The peak discharge–precipitation relationship of HBV and mHM is less straightforward than the one of SAC and VIC. HBV and mHM also show high discharge when precipitation input is high, but may in some cases still produce high discharge values even for low precipitation inputs. Such low precipitation inputs can also lead to high peak discharge for SAC but to a lesser degree than HBV and mHM.

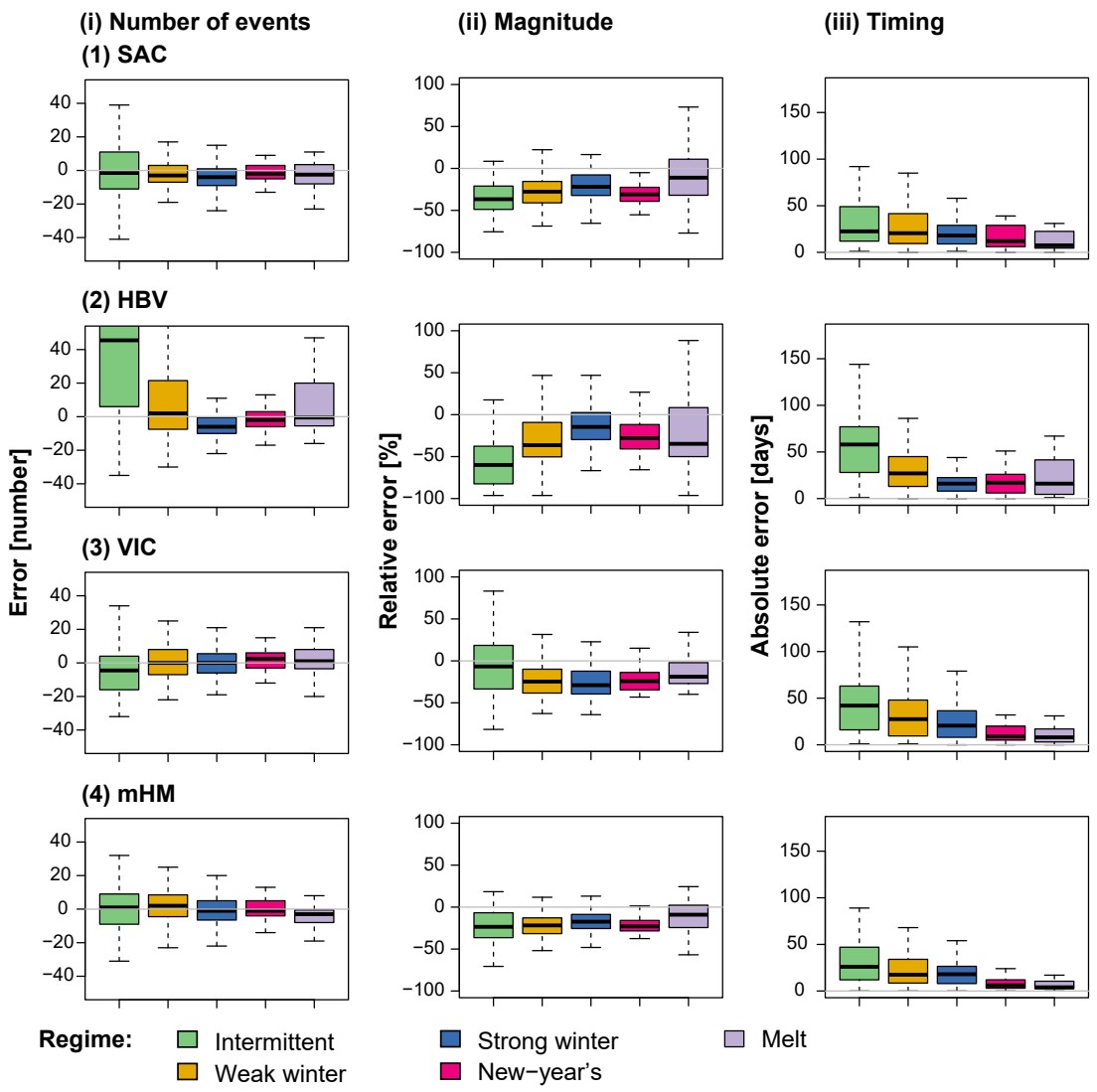

**Figure 3.** Model errors per regime type computed over the period 1981–2008: intermittent (114 catchments), weak winter (108), strong winter (176), New Year's (50), and melt (40) (Figure 1). Errors are shown for (i) number of events (error in number of events), (ii) magnitude (mean relative error), and (iii) timing (mean absolute error in days) for the four models (1) SAC, (2) HBV, (3) VIC, and (4) mHM. The boxplots are composed of one value per catchment belonging to the respective regime class.

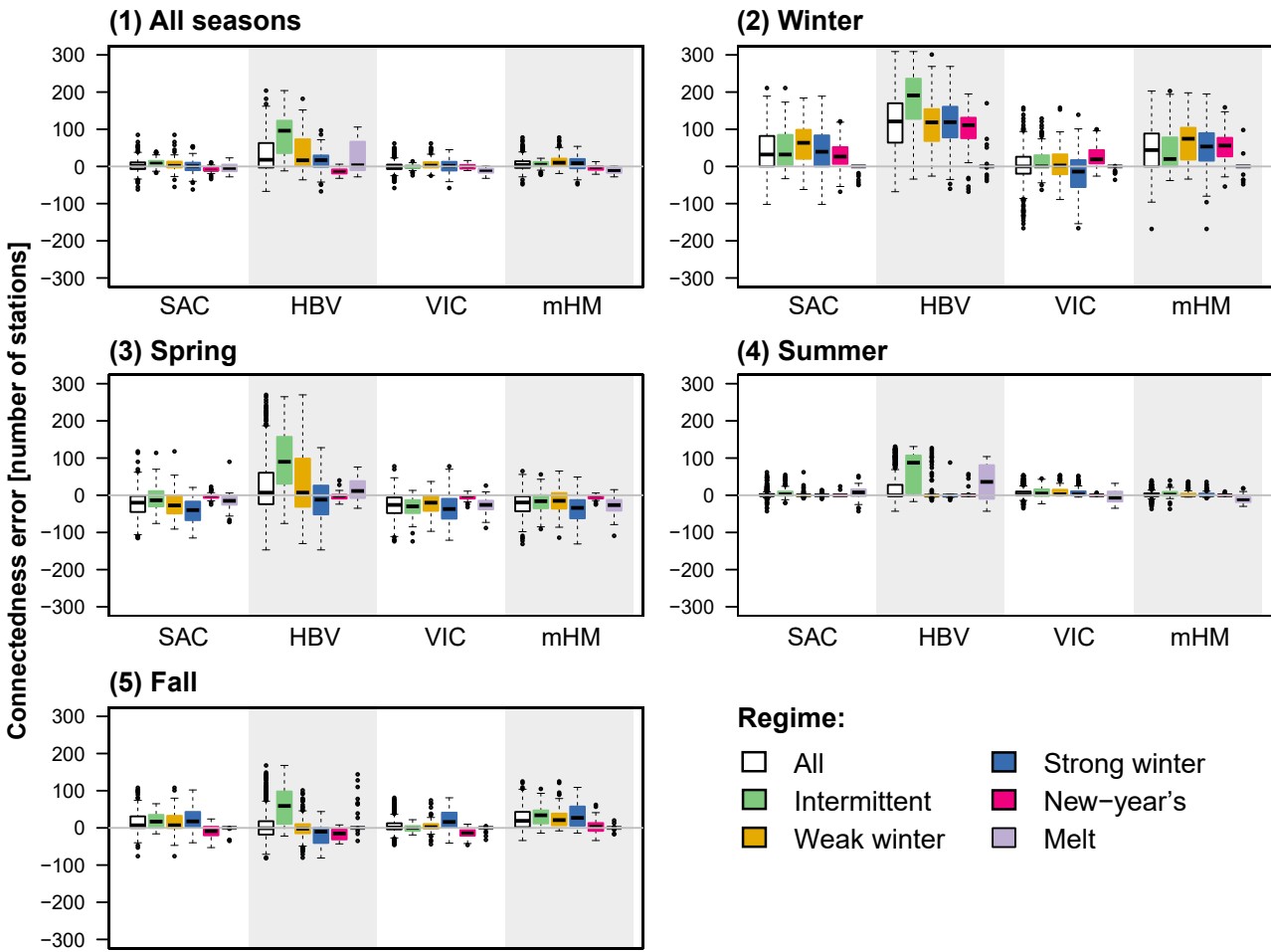

**Figure 4.** Overall (1) and seasonal (2–5) errors in flood connectedness (simulated minus observed connectedness), i.e. number of catchments a catchment is sharing at least 1% of the total number of flood events with, for the four models SAC, HBV, VIC, and mHM over all regimes and per regime: intermittent (114 catchments), weak winter (108), strong winter (176), New Year's (50), and melt (40).

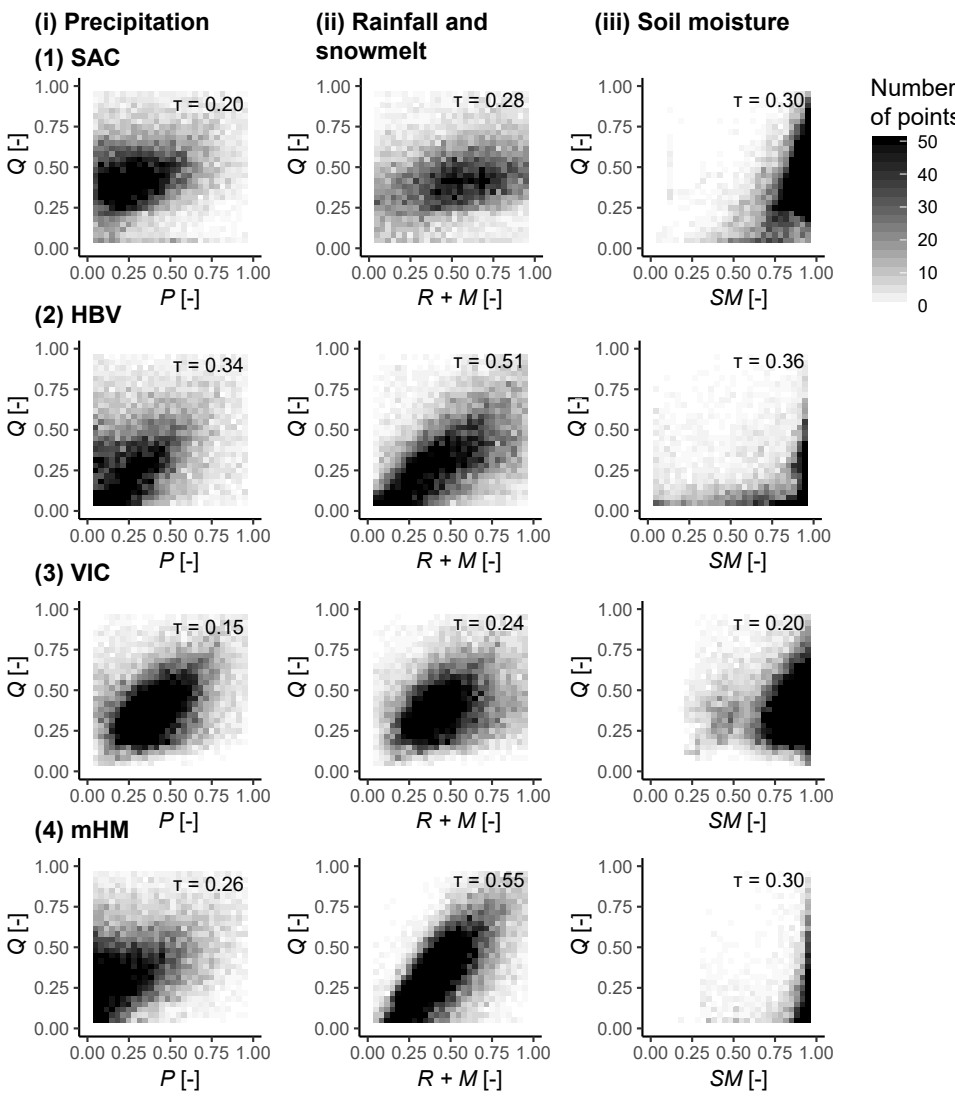

**Figure 5.** Simulated relationships between normalized flood discharge ($Q$) and normalized precipitation (i, $P$), rainfall and snowmelt (ii, $R + M$), and soil moisture (iii, $SM$, upper two soil layers for mHM) over all catchments represented by a binned scatter plot for the four hydrologic models (1) SAC, (2) HBV, (3) VIC, and (4) mHM. The darker the color, the higher the number of points within a bin (one point per catchment and event). Kendall's correlation coefficients are provided in the upper right corners of the subplots.

However, peak discharge and rainfall plus snowmelt show a strong linear relationship, i.e. the higher the combined rainfall and snowmelt input to the system, the higher is peak discharge. High flows are in most cases related to nearly full storage states but can occasionally also be triggered when soil moisture is low for SAC and VIC and to a lesser degree for HBV.

## 3.4 Floods under change

In addition to looking at how well local and spatial flood characteristics are represented by models, we look at how changes in temperature and event precipitation are translated into changes in flood flows to assess each model's suitability for climate impact assessments on floods. Our sensitivity analysis shows that the models have difficulty translating changes in event temperature and precipitation into sensitivities of flood flows (Figure 6), which can be problematic if we would like to use such models in climate change assessments. Generally, flood flows show a relatively low sensitivity to changes in mean event precipitation and temperature. This is in contrast to the behavior for mean flow, which is strongly influenced by changes in mean precipitation as demonstrated in a similar experiment by Brunner et al. (2020b). The much stronger relationship between mean precipitation and flow than between event precipitation and flow might arise because mean flow is a climate signal (Knoben et al., 2018), whereas floods are more an event (higher frequency, short-term) signal. However, some catchments, e.g. the Tucca Creek (New Year's regime) show a clear relationship between peak magnitude and both event temperature and precipitation. While these relationships are captured for some catchments (e.g. Blackwater River, weak winter regime or Tucca Creek, New Year's regime), they aren't in other catchments. The simulated sensitivities may even point in another direction than the observed ones (e.g. Pacific Creek, melt regime). In the case of melt regimes, the misrepresentation of flood sensitivities by models suggests that they may have difficulty simulating snow-influenced flooding.

This relatively poor model performance in capturing observed flood sensitivities can be generalized to the larger set of catchments studied here (Figure 7). Temperature sensitivities are found to be positive or negative, i.e. an increase in temperature could lead to an increase or decrease of peak flow depending on the catchment. In general, these temperature sensitivities are relatively weak (i.e. gradients are close to zero), which may be the reason why they are difficult to capture. In contrast, precipitation sensitivities are mostly positive, i.e. an increase in event precipitation leads to an increase in peak flow. However, the strength of these sensitivities is underestimated by all models, i.e. a change in precipitation leads to a too small change in peak flow. This underestimation of sensitivity can be understood by the underestimation of flood magnitude in general.

## 4 Discussion

### 4.1 Model performance in simulating floods

The results presented in this study demonstrate that simulating floods using hydrological models is challenging both at a local and spatial scale. At the local scale, flood timing and magnitude may not be perfectly captured which can translate into a sub-optimal representation of spatial dependencies because space and time are closely related. The challenges related to flood

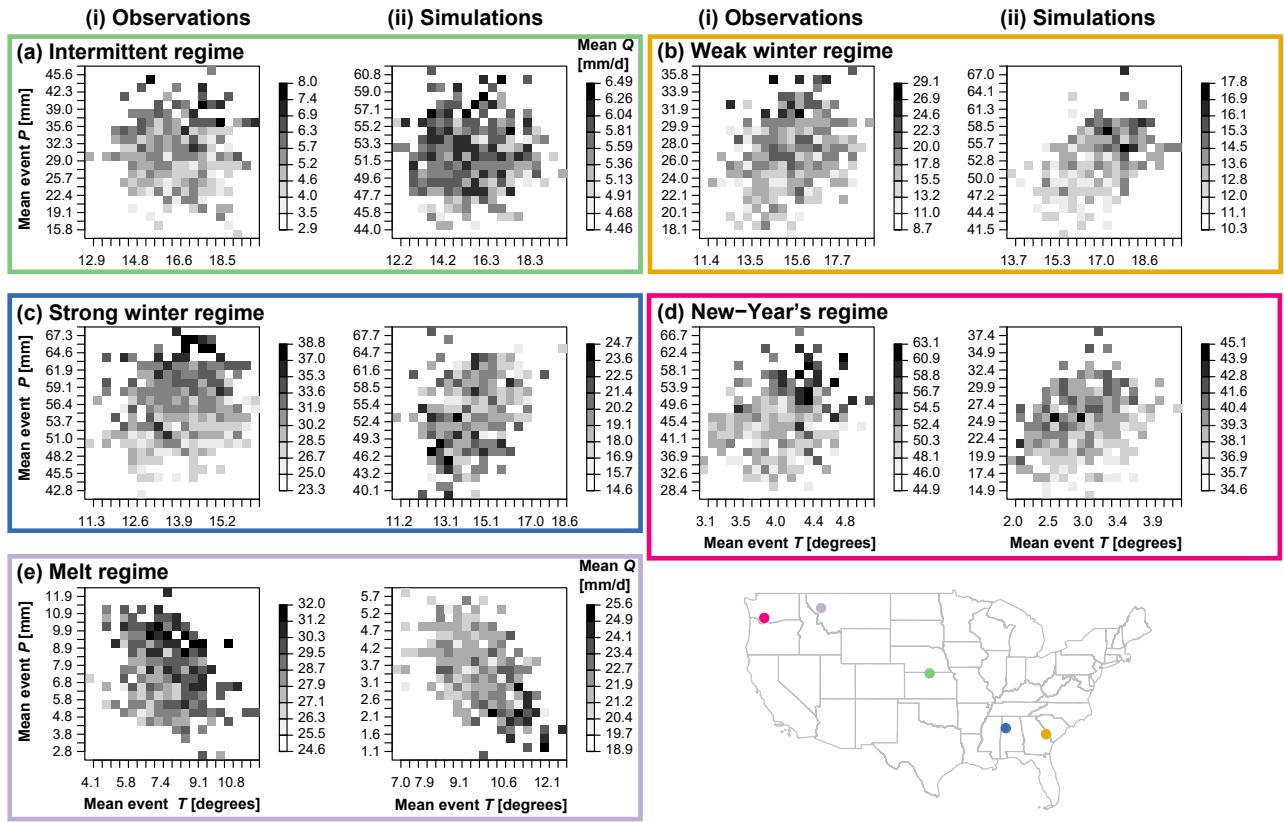

**Figure 6.** Climate sensitivity analysis for the VIC model: Dependence of mean POT magnitude ($Q$) on mean flood event precipitation (1-day; $P$) and mean flood temperature ($T$) for five example catchments, those with the best $E_{\text{KG}}$ per regime type: intermittent regime (green; USGS ID 09210500 Fontanelle Creek near Fontanelle, WY; $E_{\text{KG}} = 0.78$), weak winter regime (yellow; USGS ID 02369800 Blackwater River near Bradley, AL; $E_{\text{KG}} = 0.83$), strong winter regime (blue; USGS ID 11522500 Salmon River above Somes, CA; $E_{\text{KG}} = 0.84$), New Year's regime (pink; USGS ID 14303200 Tucca Creek near Blaine, OR; $E_{\text{KG}} = 0.9$), and melt regime (purple; USGS ID 13011500 Pacific Creek at Moran, WY; $E_{\text{KG}} = 0.92$). Grid axes and grey scales differ between plots where darker colors indicate higher flood magnitudes.

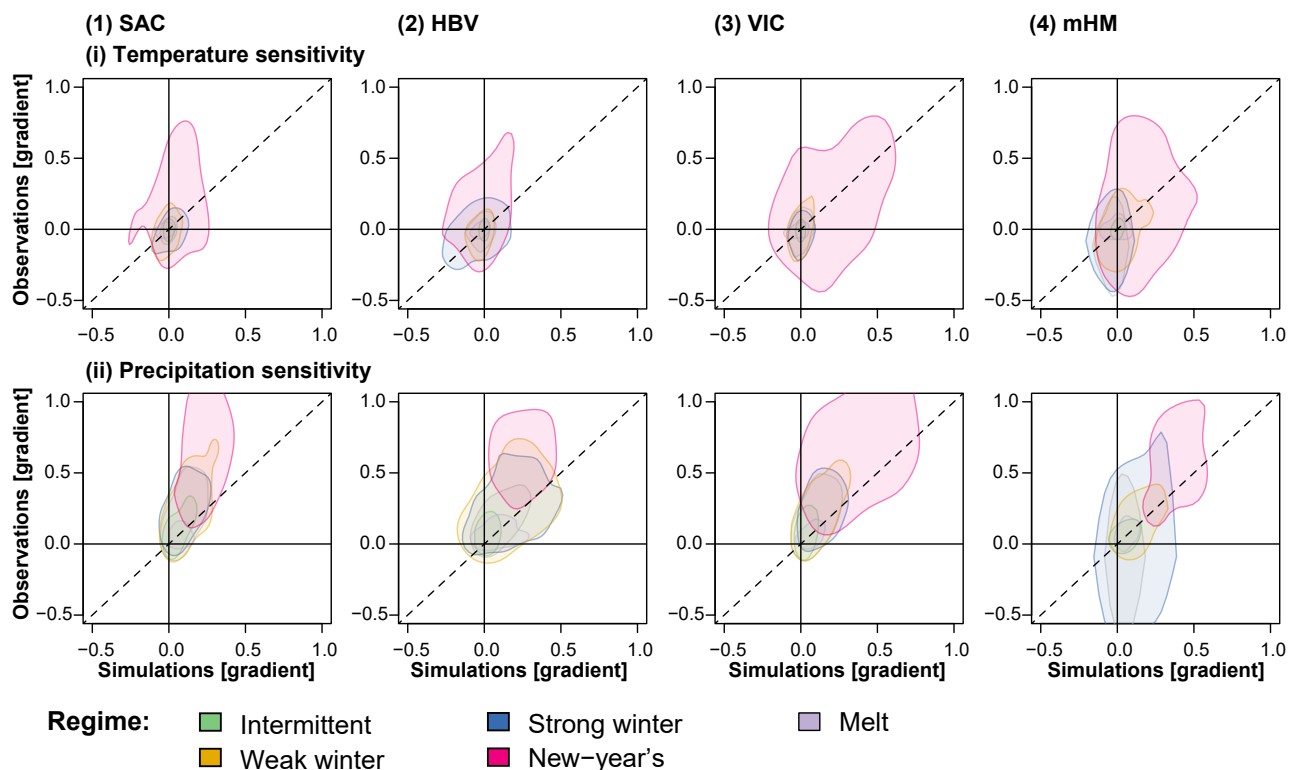

**Figure 7.** Observed vs. simulated (i) horizontal (temperature) and (ii) vertical (precipitation) climate sensitivities for floods represented by two-dimensional kernel density estimates for the four models (1) SAC, (2) HBV, (3) VIC, and (4) mHM for the five regime types: intermittent (114 catchments), weak winter (108), strong winter (176), New Year's (50), and melt (40) (Figure 1). Positive and negative values indicate positive and negative associations of precipitation and temperature with peak flow, respectively. Values on the dashed line indicate correspondence between observed and modeled sensitivity gradients.

simulations become especially pronounced under climate conditions different from the current ones because additional sources of uncertainty are added to the modeling chain.

Even though the models have been calibrated for the local situation, substantial differences in magnitude and timing were found between observations and simulations. Locally, simulated floods showed smaller magnitudes and had different timing than observed ones while the number of floods was reproduced relatively well except by the HBV model for catchments with intermittent regimes. The flood magnitude underestimation found for all four models tested is in line with previous studies showing that using $E_{KG}$ individually results in an underestimation of peak flow (Mizukami et al., 2019) due to an
underestimation of variability, which will result in an under-representation of extremes (Katz and Brown, 1992). Another factor potentially contributing to this underestimation is that the models were forced with spatially lumped instead of distributed data, which may have smoothed the simulated discharge response.

Under the current calibration paradigm, where models are calibrated to local discharge conditions using $E_{KG}$ as objective function, flood conectedness is not accounted for. As a result, flood contectedness is not well captured by the models as illustrated by the finding that flood connectedness is over- or underestimated depending on the season. The overestimation of spatial dependence in winter for all regimes except the melt regime is likely related to higher simulated than observed snowmelt as high soil moisture and snow availability have been shown to increase spatial flood connectedness (Brunner et al., 2020a). Related to this, the underestimation of spatial connectedness in spring may be related to the subsequent missing snowmelt contributions. Spatial connectedness in summer has been shown to be generally weak due to the occurrence of localized, convective events (Brunner et al., 2020a), which is reflected by most models except for HBV in the case of intermittent and melt regimes. Spatial flood connectedness has also been shown to be weak in fall (Brunner et al., 2020a) but is overestimated by most models. The finding that there is room to improve the representation of spatial flood dependencies is in line with previous studies showing that large-scale hydrological models have a weakness in reproducing regional aspects of floods (Prudhomme et al., 2011).

There are slight variations in performance among models. These variations may result from differences in the representation of flood producing mechanisms as indicated by distinct behaviors in how the models translate precipitation into runoff. VIC and SAC show more linearity in their event precipitation and peak discharge relationship than HBV and mHM, possibly because VIC and SAC have the capability to generate surface runoff when precipitation intensity exceeds infiltration capacity (Burnash et al., 1973; Liang et al., 1994). In this case, incoming precipitation is directly translated into flood discharge. In contrast, HBV and mHM, the latter which is based on the HBV model structure (Kumar et al., 2013), does not include a surface runoff component and all discharge originates in the model stores (Bergström, 1976). This introduces a non-linearity in model response and may explain why a smaller precipitation input may still generate high peak flows in these models. These differences in process representation suggests that a 'most suitable model' could be identified for a specific application at hand. If one is e.g. interested in simulating floods in catchments with intermittent regimes, the HBV model does not seem to be an ideal choice because it there simulates too many floods with a too small magnitude. The overestimation of the number of events in catchments with intermittent regimes by HBV may be explained by its fast response to precipitation as expressed through its model parameter $\beta$, which introduces non-linearity to the system (Viglione and Parajka, 2020).

Our climate sensitivity analysis shows that the simulation of floods becomes even more challenging under climate conditions different from the current ones as the hydrological models employed in this study have limited capability in reproducing observed hydrologic sensitivies during flooding. These limitations may be related to input uncertainties (Te Linde et al., 2007), insufficient model calibration (Fowler et al., 2016), or equifinality in process contributions for simulations with (very) similar efficiency scores, leading to an inability to unambiguously identify the appropriate relative process contributions (Khatami et al., 2019).

## 4.2 Potential ways to improve model performance

The results of our model comparison highlight that there is room for improvement regarding the representation of local flood events, spatial flood dependence, and flood producing mechanisms. We here discuss four potential ways for improving model performance: developing flood-tailored calibration metrics, considering spatial aspects in model calibration, improving representation of flood processes, and representing input uncertainties.

A first possibility to improve model performance is to develop calibration metrics tailored to flooding instead of relying on $E_{KG}$. Our results show that $E_{KG}$ can lead to simulation performance deficits for phenomena of interest, including an underestimation of peak flow, a misrepresentation of timing, and over- or underestimation of seasonal spatial flood connectedness. As is evident in some existing practice-oriented applications of hydrological models (Hogue et al., 2000; Unduche et al., 2018; World Meteorological Organization, 2011), the simulation of floods and other hydrologic phenomena is likely to be improved by using more tailored model calibration strategies. The representation of streamflow variability could potentially be improved by giving more weight to the variability component of an integrative metric such as the $E_{KG}$ (Pool et al., 2017); whereas the representation of flood magnitude and timing may be improved by giving more weight to the bias and correlation components of the $E_{KG}$. Alternatively, these characteristics could be optimized explicitly by minimizing error in key hydrograph signatures related to site-specific flood phenomena. Such flood-focused optimization may similarly to $E_{KG}$ rely on multiple objectives in a scalar function (Gupta et al., 1998; Efstratiadis and Koutsoyiannis, 2010) such as volume error, root-mean-squared error, and peak flow error (Moussa and Chahinian, 2009); $E_{NS}$ and relative peak deviation (Krauße et al., 2012); $E_{KG}$, peak efficiency, and logarithmic efficiency (Sikorska et al., 2018); or $E_{KG}$, peak efficiency, and mean absolute relative error (Sikorska-Senoner et al., 2020). In addition, model performance can potentially be improved by using multiple metrics describing important catchment processes (Madsen, 2003; Dembélé et al., 2020), i.e. flood generating mechanisms such as soil moisture and snowmelt.

A second way to improve model performance is to focus on the spatial representation of extremes, which may be improved by considering spatially distributed features of model response or spatial correlation within a spatial calibration framework. Such a framework could build upon existing spatial verification metrics such as the spatial prediction comparison test used e.g. to validate precipitation forecasts (SPCT; Gilleland, 2013), Empirical Orthogonal Functions (EOFs), or Kappa statistics (Koch et al., 2015). For the calibration and evaluation of spatially-distributed hydrological models, Koch et al. (2018) recently proposed the SPAtial Efficiency (SPAEF) metric which reflects three equally weighted components: correlation, coefficient of variation and histogram overlap. To improve the spatial dependence of floods across different sites, such spatial calibration frameworks would need to include spatial verification metrics focusing at extremes, which could e.g. be achieved by looking at deviations of simulated from observed F-madograms, which measure extremal dependence (Cooley et al., 2012). Please note, however, that even the use of spatial verification metrics may not overcome the lack of spatial heterogeneity in precipitation or soil moisture data.

A third way of improving model performance is to test whether a model is fit-for-purpose and to identify model structures which accurately represent relevant flood producing mechanisms. The importance of model structure choice has been highlighted in previous studies both for low- and high flow events (Melsen and Guse, 2019; Kempen et al., 2020; Knoben et al.,

2020) and should depend on the spatial complexity of the phenomenon studied (Hrachowitz and Clark, 2017). However, model structure choice for a specific application is not straightforward and automatic model structure identification frameworks have only been introduced very recently (Spieler et al., 2020). To improve the representation of flood processes, such frameworks would ideally explicitly consider local and spatial flood characteristics and the representation of different flood generation processes such as rain-on-snow events or flash floods. The representation of rain-on-snow floods for example requires an accurate representation of the energy balance in order to represent factors affecting snowmelt processes such as net radiation and turbulent heat fluxes (Pomeroy et al., 2016; Li et al., 2019) .

A fourth possibility to improve model performance is to address data uncertainty of streamflow observations and of precipitation input. Errors in streamflow measurements caused by stage-discharge rating-curve uncertainty (Coxon et al., 2015; Kiang et al., 2018) influence model calibration and evaluation. To improve uncertainty estimates, such uncertainty should be accounted for by explicitly considering streamflow measurement uncertainty in model calibration (McMillan et al., 2010). In addition, the uncertainty of the precipitation product used to drive a hydrological model can lead to differences in observed and simulated flows (Te Linde et al., 2007; Renard et al., 2011). Precipitation products may show observation uncertainties (Mcmillan et al., 2012) and underestimate extreme rainfall or the spatial dependence of extreme precipitation at different locations because spatial smoothing or averaging during the gridding process reduces variability (Haylock et al., 2008; Risser et al., 2019). Such spatial uncertainty could be accounted for by using probabilistic analyses of precipitation fields (Newman et al., 2015a; Frei and Isotta, 2019). The consideration of such input uncertainty is particularly important if we are interested in future changes because of climate model and scenario uncertainty, where precipitation uncertainty is specifically pronounced (Chen et al., 2014; Lopez-Cantu et al., 2020). Even though many of these possibilities have been discussed in previous studies, their consideration in flood analyzes is not a standard practice.

## 5 Conclusions

Our model comparison shows that flood characteristics are not always well captured in hydrological models developed for research studies – even when the models have been calibrated with a calibration metric perceived suitable for flood modeling, the Kling–Gupta efficiency metric ($E_{KG}$). The number of flood events were over- or underestimated depending on the catchment, flood magnitudes were underestimated by all models in most catchments, and the ability of the model to accurately reproduce event timing was proportional to the hydroclimatic seasonality. These model deficiencies in reproducing local flood characteristics, especially timing, can lead to a misrepresentation of spatial flood dependencies, particularly in winter, because the temporal and spatial dimension of flooding are closely linked. Our sensitivity analysis also shows that climate sensitivities of floods, especially to changes in precipitation, are not well represented in models even if the model can be deemed 'well-calibrated' according to the $E_{KG}$ metric. These sensitivities are generally underestimated by models independent of the geographical areas considered, i.e. an increase in event precipitation may not be translated into a strong enough increase in flood peak. The mis-estimation of these sensitivities may undermine the reliability of future flood hazard assessments relying on such models.

The limited capability of the models in reproducing local and spatial flood characteristics and the sensitivity of runoff to precipitation inputs is partly attributed to model structure and partly to a reliance of the calibration on an individual variable (streamflow) and metric ($E_{KG}$). While $E_{KG}$ is integrative of certain properties (bias, variance, correlation), it does nonetheless not explicitly focus on high flow values, their spatial dependencies, or processes generating high flow values. We conclude that calibration using only an individual model performance metric or variable can result in model implementations that have limited value for specific model applications, such as local and in particular spatial flood hazard analyses and change impact assessments. This study underscores the importance of improving the representation of magnitude, timing, spatial connect-edness, and flood generating processes. Potential ways of achieving such improvements include developing flood-focused, multi-objective and spatial calibration metrics, improving flood generating process representations through model structure comparisons, and reducing uncertainty in precipitation input. Such steps are recommended to improve the reliability of flood simulations and ultimately local and regional flood hazard assessments under both current and future climate conditions.

*Data availability.* Observed streamflow measurements were made accessible by the USGS and can be downloaded via the website https://waterdata.usgs.gov/nwis. Simulated streamflow, precipitation, and storage time series can be requested from Lieke Melsen (lieke.melsen@wur.nl) for the SAC, HBV, and VIC models and for the mHM model from Oldrich Rakovec (oldrich.rakovec@ufz.de).

## Appendix A: Model illustrations

This section provides illustrations of the model structures used in this work. Model schematics summarize the model states and fluxes. Schematics and equations use model-specific names as they are used in the model code. For clarity, these descriptions enforce that fluxes are shown in lower case and states in upper case. The model diagrams are based on:

– Snow17/SAC-SMA: analysis of the model's description (National Weather Service NOAA, 2002): https://www.nws.noaa.gov/oh/hrl/general/chps/Models/Sacramento_Soil_Moisture_Accounting.pdf and source code.

– TUW HBV: analysis of the model's source code (Viglione and Parajka, 2020).

– VIC: descriptions of VIC in Melsen et al. (2018); Melsen and Guse (2019) and on analysis of the v4.1.2h source code (https://github.com/UW-Hydro/VIC/releases/tag/VIC.4.1.2.h).

– mHM: analysis of the model's source code (https://git.ufz.de/mhm/mhm/-/tree/5.7) and a diagram provided in (Kumar et al., 2010).

## A1 Snow17/SAC-SMA

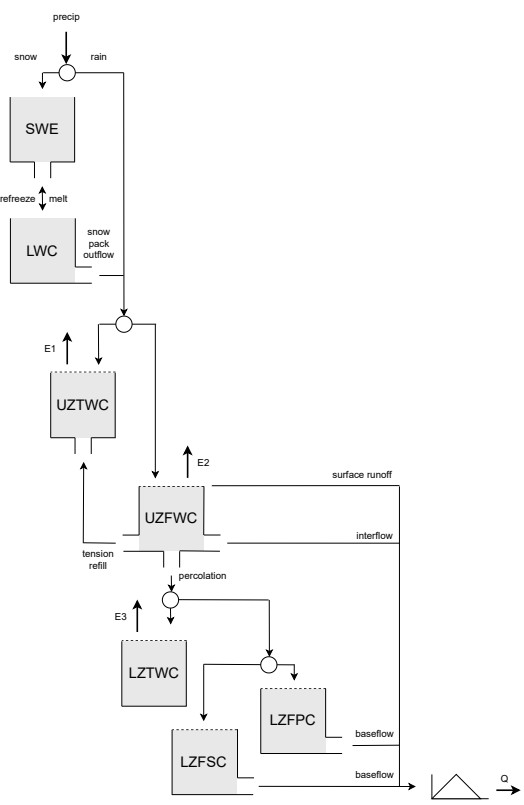

**Figure A1.** Structure of the Snow17/SAC-SMA model. Fluxes: precipitation (precip), snow, rain, snowmelt (melt), refreeze, snowpack outflow, evapotranspiration (E1, E2, and E3), tension refill, surface runoff, interflow percolation, baseflow, simulated discharge (Q). States: snow-water-equivalent (SWE), liquid water content (LWC), upper zone tension water contents (UZTWC), upper zone free water contents (UZFWC), lower zone tension water contents (LZTWC), lower zone free primary contents (LZFPC), lower zone free supplemental contents (LZFSC).

## A2 TUW-HBV

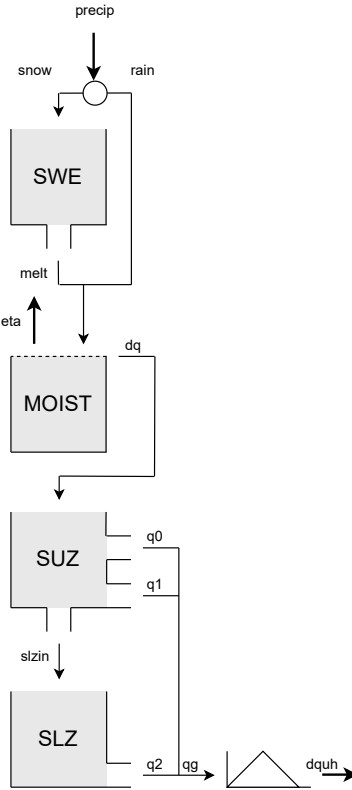

**Figure A2.** Structure of the TUW HBV model. Fluxes: precipitation (precip), snow, rain, snowmelt (melt), actual evapotranspiration (eta), runoff (dq), surface runoff (q0), subsurface runoff (q1), baseflow (q2), simulated runoff (qg), simulated discharge (dquh), input from upper to lower storage (slzin). States: snow-water-equivalent (SWE), soil moisture (MOIST), upper storage zone (SUZ), lower storage zone (SLZ).

## A3 VIC

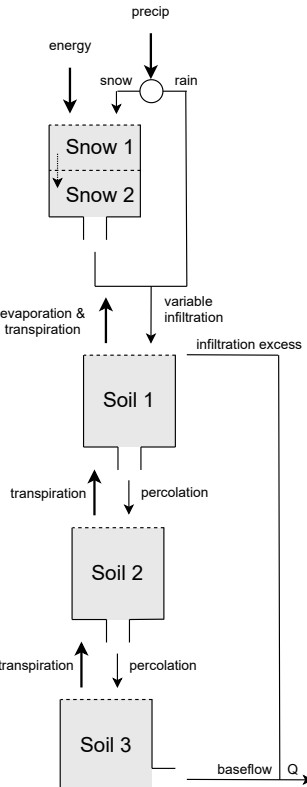

**Figure A3.** Fluxes: precipitation (precip), energy, snow, rain, variable infiltration, evaporation and transpiration, infiltration excess, baseflow, percolation, transpiration, simulated runoff (Q). Storage: snow layer 1 (Snow 1), snow layer 2 (Snow 2), soil layer 1 (Soil 1), soil layer 2 (Soil 2), soil layer 3 (Soil 3).

 **A4** **mHM**

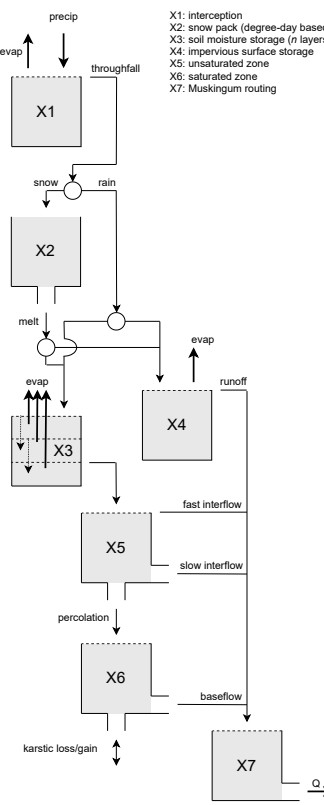

**Figure A4.** Fluxes: precipitation (precip), evapotranspiration (evap), throughfall, snow, rain, snowmelt (melt), runoff, fast interflow, slow interflow, percolation, baseflow, karstic loss/gain, simulated discharge (Q). Storage: Interception storage (X1), snow pack (X2), soil moisture storage (X3), impervious surface storage (X4), unsaturated zone (X5), saturated zone (X6), routing (X7).

*Author contributions.* MIB and MPC developed the study design. NM, OR, and LAM provided the model simulations and together with MIB, MPC and WK interpreted the model output. AW assisted with the paper's background and messaging and proposed the climate sensitivity strategy. WK produced the model illustrations. MIB wrote the first draft of the manuscript and all co-authors revised and edited the manuscript.

*Competing interests.* The authors declare that they have no conflict of interest.

*Acknowledgements.* This work was supported by the Swiss National Science Foundation via a PostDoc.Mobility grant (P400P2_183844, granted to MIB). We acknowledge co-author support by the Bureau of Reclamation (CA R16AC00039), the US Army Corps of Engineers (CSA 1254557), and the NASA Advanced Information Systems Technology program (award ID 80NSSC17K0541). We also acknowledge support from the Global Water Futures research programme. We thank the editor and the four reviewers for their constructive feedback, which helped to reframe and clarify the storyline.

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
