# Peer review of "Flood spatial coherence, triggers and performance in hydrological simulations: large-sample evaluation of four streamflow-calibrated models."

_Hydrology and Earth System Sciences, 2020_

## Referee Comment (RC1) · Anonymous Referee #1 · 26 May 2020

General Comments

This paper assesses how well hydrological models calibrated using the Kling-Gupta Efficiency (KGE) metric can reproduce local and regional flood characteristics. Streamflow simulations from four hydrological models are evaluated across a large sample of hydrologically varied catchments, for flood timing, magnitude and spatial variability. In addition, the authors explore the model sensitivities of high flows to precipitation and temperature. This is an interesting analysis and helps to explain model deficiencies for hazard and change impact assessments.

I enjoyed reading this paper, which is well-written, concise and easy to follow. The

figures are all relevant and well-presented. My main concern is that the title, and focus on deficiencies of integrated calibration metrics, does not accurately reflect the study. I think the title suggests that different model calibration strategies are going to be implemented and evaluated, or that there is going to be some assessment of performance for different calibration strategies. The study only looks at models calibrated using KGE, and it is therefore hard to distinguish if any failure of the models in representing flood characteristics is due to the calibration strategy or other factors such as quality of input and observed streamflow data, or model structural errors.

Overall, I think this paper would make an interesting contribution for HESS, following changes and clarifications to the manuscript. I have several specific comments which I outline below.

Specific comments

Title: as discussed in general comments, I am not sure the title best reflects the content of the paper. I think this title suggests evaluation of different calibration strategies, whereas KGE has been used throughout. A title focusing on key results/ what has been done (e.g. Evaluating hydrological model suitability for flood impact assessments across a large sample of catchments) may be more suitable.

Line 10: "Our results show that both the modelling of local and spatial flood characteristics is challenging." It could be helpful to highlight some the key results in the abstract to justify this statement, i.e. all models under predict the magnitude of events.

Line 12: "We conclude. . ." The manuscript focuses on models calibrated on KGE alone, and infers that deficiencies in model performance is due to the model calibration. I think this is quite a big leap – as there are other factors which could result in poor model performance (e.g. errors in observed precipitation and river flow data, particularly for peak flow events). It would be good to discuss these within the manuscript.

Introduction:

Line 25: There is a tendency for high values to be underestimated and low values to be overestimated (Gupta et al. 2009), but I am not sure it is correct to say that the optimal value actually underestimates flow variability. It could be worth mentioning that NSE is often used for high flow studies, based on the idea that by using squared errors it mostly constrains peaks and high flows (Mizukami et al. 2019).

Line 33: I do not completely follow this sentence – why non-flood-related signature?

Data and Methods: Whilst the methods section is clear overall, I felt that a few sections needed clarifying.

Line 68: It would be useful to add references for the models.

Line 68: It would be helpful to know some more about the differences/similarities between the models. In particular, any differences in modelling decisions that may contribute to the performance differences (it would be good to explain why HBV does so poorly compared to the other models). Perhaps a table of key differences or a figure giving model structure diagrams would be helpful.

Line 70: "model parameters were calibrated on streamflow observations by minimising the EKG" – How was the optimisation performed (e.g. which algorithm was used) and is this the same in both studies? Was mHM calibrated using multiscale parameter regionalisation, and if so was EKG evaluated across the region rather than for each catchment? It would be useful to know how the calibration differed, despite all being based on KGE.

Line 80: How do these meteorological forcing data differ? Are they both the same timestep?

Line 67-83: The dates used for the simulations are unclear. In the method a few different date ranges are given: Line 67: "we use daily streamflow simulations for the period 1981-2008", Line 82: "SAC, HBV and VIC were evaluated on the period 1985-2008", "mHM was calibrated on the period 1999-2008 and evaluated on the period

1989-1999." It seems that 1981-1985 were not used in the previous studies. It would be useful to know which period the model simulations were actually run for, whether a warm-up period has been given, and how long the warmup was. Also, over which period were SAC, HBV and VIC calibrated? Does the period 1981-2008 refer to hydrological years or calendar years? It would be helpful to give months here.

Line 85: Have you used the KGE values given by Mizukami et al. (2019) and Melsen et al. (2018), or were these re-calculated these over the period 1981-2008? I assumed all model performance was calculated over the same period, against the same observed discharge data, but this is not clear. Line 85: I agree that performance is generally lowest for catchments with intermittent regimes, but there is a lot of overlap in performance.

Line 114: "we then use the data sets resulting from Step 2 to evaluate how models reproduce overall and seasonal spatial flood dependence." It would be useful to have a bit more detail in this section. How was the error statistic calculated?

Line 117: It is not clear if 1% was the value used. This should be made clear, and it would help to have a reference/justification for why this value was chosen.

Line 122: "Time of concentration is typically less than one day for small headwater basins." This needs a reference.

Results: A key advantage of this study is the application of multiple model structures to a large sample of catchments. Throughout the methods/results it would be useful to have more discussion of the differences between the models. In particular, it would be useful to know why HBV performs so poorly compared to the other models for flood magnitudes.

Line 145: "For most catchments, the number of flood events is relatively well simulated by most models..." It would be useful to know the number of observed events, to put these errors into context. I am assuming that the number of events is similar between

all regime types due to the selection of the threshold. Otherwise a percentage error may be easier to interpret.

Line 150: Underestimation of peak flow is attributed to the KGE metric underestimating variability, and spatially lumped model inputs. This could also be due to data errors – for example, McMillan et al. (2012) show that there can be large uncertainties associated with precipitation products. It would be useful to add this to the discussion. McMillan, H., Krueger, T., & Freer, J. (2012). Benchmarking observational uncertainties for hydrology: rainfall, river discharge and water quality. Hydrological Processes, 26(26), 4078-4111.

Line 152: "the use of lumped forcings may also artificially synchronize hydrologic response, which would lead to overestimation." – could this be further explained?

Line 163: "the overestimation of spatial dependence in winter is likely related to higher simulated than observed snowmelt." I was not sure which regime(s) this comment was referring to. The melt regime is the only one that doesn't show an overestimation of spatial dependence in winter for any model.

Line 170: "Connectedness overestimation is most pronounced..." I don't agree with this sentence. For the other 3 models intermittent regime does not seem to be overestimated any more than other regime types. In winter, it seems to be in line with and below all regimes for SAC, VIC and mHM.

Line 177: "There is a clear positive relationship ..." I would not say there is a clear positive relationship for SAC. Perhaps a slight positive relationship.

Line 180: "soil moisture and event magnitude are also positively related..." I would interpret this a little differently for VIC - at full saturation we see events of all magnitudes. It is just the upper level of flows that is increasing with soil moisture. 'lower values' - does this mean lower values of peak Q?

Line 183: I think this is also the case for SAC.

Line 186: "for SAC and VIC." I would add that to some extent this is also the case for HBV.

Line 199: Why is this the case?

Line 211: "These relationships are, however, not necessarily captured by the models..." It may be worth highlighting that in some areas these relationships are generally captured by the models: e.g. weak winter regime broadly captures the precipitation relationship, and New-Year's regime captures precipitation and temperature relationship.

Figure 6: This figure has a lot of text, which can be distracting from the plots. I think it would be help to simplify the y axes and colorbar scales to 2 significant figures (i.e. no decimal places).

Figure 6: It would be clearer if the colour scales matched between the observations and simulations for a specific catchment, and also the x and y axis ranges. Otherwise, it would be useful to point out that the scales differ, and explain why this has been done, i.e. colours ranging from the largest to smallest flood.

Line 221: I do not follow this link -could this be explained better? It feels like there is a jump from models inadequately representing the sensitivity of peak flows to precipitation to errors in precipitation data being the cause.

Line 223: ".. may be related to insufficient model calibration..." This feels like quite a big leap. Having only looked at models calibrated using KGE it doesn't feel like there is enough information to attribute poor performance to calibration metrics. Could it be the model structures more generally, or the input data errors, that are causing these model deficiencies rather than the calibration metric?

Figure 7: It would be helpful to have a more thorough explanation of this figure. Perhaps a sentence explaining that positive values mean an increase in the variable leads to an increase in peak flows, and values falling on the dotted line indicate simulations match

observations.

Line 226: This sentence implies an underestimation in timing. Only absolute errors in day of flood timing are given, not the direction of change within the year. Maybe rephrase this sentence.

Conclusions:

Line 235: In the introduction a key aim is 'assess which aspects of hydrological models may need to be improved ....' and 'identifying and documenting model weaknesses regarding regional and future flooding will highlight advances for future model development.' .. These aims/questions could be more directly addressed in the conclusions section.

Technical corrections

Line 86: "successfully" should be "success"

Line 160: "underestimates" should be "underestimate"

HESS Review Checklist

In the full review and interactive discussion, the referees and other interested members of the scientific community are asked to take into account all of the following aspects:

1) Does the paper address relevant scientific questions within the scope of HESS? YES

2) Does the paper present novel concepts, ideas, tools, or data? YES

3) Are substantial conclusions reached? YES

4) Are the scientific methods and assumptions valid and clearly outlined? YES

5) Are the results sufficient to support the interpretations and conclusions? MOSTLY

6) Is the description of experiments and calculations sufficiently complete and precise

to allow their reproduction by fellow scientists (traceability of results)? YES

7) Do the authors give proper credit to related work and clearly indicate their own new/original contribution? YES

8) Does the title clearly reflect the contents of the paper? NO

9) Does the abstract provide a concise and complete summary? YES

10) Is the overall presentation well structured and clear? YES

11) Is the language fluent and precise? YES

12) Are mathematical formulae, symbols, abbreviations, and units correctly defined and used? YES

13) Should any parts of the paper (text, formulae, figures, tables) be clarified, reduced, combined, or eliminated? MINOR CLARIFICATIONS TO METHODS

14) Are the number and quality of references appropriate? YES

15) Is the amount and quality of supplementary material appropriate? YES

---

## Referee Comment (RC2) · Anonymous Referee #2 · 13 Jun 2020

This is a well-written journal with appropriate content for HESS. I think this is a nice study, though I do have some suggestions related to the framing of the work and its discussion. This could be a very nice paper if the focus was actually on the calibration strategy.

My comments are:

[1] The title of the study suggests a wide-ranging assessment of different calibration strategies in the context of flood modelling. However, the study is essentially an assessment of the value of using KGE for flood modelling. The actual focus is fine, but I think it should be reflected in the title of the manuscript to avoid confusion.

[Figure]

[2] Given that the focus of the manuscript is on the calibration strategy, I was surprised to not find any details on what strategy was used to find the best KGE values? What algorithm was used etc would be helpful information for the reader to understand what has been done. While this might be covered in previous papers in detail, it would be good to see at least some basic description here as well.

[3] It would also be helpful to have some calibration/validation results for each model to distinguish them at this point already (if they differ?).

[4] Section 3.1: Why is HBV so poor? Especially given its focus on snow/cold regions?

[5] Section 3.1: I am a bit confused by this assessment. Are you assessing the model or the metric used for calibration? The paper title suggests that the focus is on the calibration strategy, so my question is why using the same calibration strategy results in different model performance? Significant differences between very similar models is surprising if the models have been calibrated in the same manner.

[6] Lines 224-225: But how do you know that if you only assessed one metric? The authors do a very nice job of including multiple models, but if the focus is on the calibration strategy, then why do you not include variability in how they calibrate the models? How can you make conclusions about the calibration strategy if you did not vary it. Would putting more weight on fitting the variability have produced a better fit to variability (using a weighted KGE)? You have this as a discussion point, but why is this not part of your actual study?

[7] Line 236: But how do you know that? Maybe all the models have the same problem regardless of calibration metric used? Maybe you did not look hard enough for an optimum parameter set?

[8] Line 245: As stated above, I find it dissatisfying to make such a conclusion. Testing this suggestion is a very minor effort given the work already presented in this paper. Why can the authors not try this? This – to me – would be part of the main tests the

authors should have done in this paper. You cannot test the implications of choices about the calibration strategy if you do not test different choices. Using multiple models does not compensate for this omission.

---

## Referee Comment (RC3) · Anonymous Referee #3 · 2 Jul 2020

Though I agree with the title of the manuscript and with the main conclusions on l. 254-259 (see below), there are several serious deficiencies in the approach and interpretation of results. The study looks like an initial stage only: let us calibrate four models for many relatively small catchments in USA using one metric, EKG, to see, how well the models will reproduce local flood characteristics and spatial aspects of flooding, and how well would they be prepared for climate impact assessment. The conclusion is that the models calibrated on the Kling–Gupta efficiency alone have limited reliability in flood hazard assessments.

Such "negative" result could be expected, as there are several recent publications

pointing on a necessity of comprehensive approaches for hydrological model calibration and evaluation (for mean flow and for extremes) and especially if they are intended for climate impact assessment (see e.g. Choi and Beven, 2007, Coron et al., 2012, Refsgaard et al., 2013, Thirel et al., 2015, Krysanova et al., 2018). Therefore, such an "initial stage" of the study should be supplemented by application of an extended approach: for example, including at least some of the further steps suggested in the papers listed above, like multi-site and multi-variable calibration (mentioned in the manuscript), DSS test checking for contrasting climate sub-periods, testing specifically for indicators of interest, i.e. for high flows and floods. Then the study would be much more valuable.

There are also other deficiencies in the applied approach and in the interpretation of the obtained results. Therefore, the manuscript should be rejected in its present form.

Other major concerns:

APPROACH

l. 81-82: were driven with Daymet meteorological forcing (Thornton et al., 2012) and mHM with the forcing by Maurer et al. (2002): → how are they comparable with the observed climate? Was the comparison done or not? If not, it would be reasonable to do.

l. 82: SAC, HBV, and VIC were evaluated on the period 1985–2008: → and calibrated for which period?

l. 110-112: → would be good to express the relative error in %, and define thresholds for acceptable performance (e.g. based on literature) for all 3 indicators. For example, is a relative error of 25% acceptable or not? The thresholds could be shown in Fig. 3 by horizontal lines to enable distinguishing the good/acceptable and poor performances.

Sec. 3.1: → to discuss performance based on the pre-defined thresholds

l. 116-118: a catchment is connected to another catchment if they share a certain

number of events, i.e. at least 1% of the total or seasonal number of events: → is 1% of shared events really sufficient to define their connectivity??? Due to that, the whole section 3.2 is questionable.

l. 127: we generate surrogate time series of temperature, precipitation, and streamflow for each catchment by resampling the available hydrological years with replacement: → the procedure is not quite clear, and should be better explained!

l. 202-203: "to assess each model' suitability for climate impact assessments on floods": → how the resampling could help to assess suitability? It would be better to test for contrasting climate subperiods, or to compare trends in discharge, high flows and POT series.

Sec. 3.3 and Fig.5: → Maybe to add correlation coefficients to better characterize the relationships?

INTERPRETATION

l. 145-146: "For most catchments, the number of flood events is relatively well simulated by most models": → this is not evident, if a threshold is not defined. It is only visible that medians are close to zero for three models, and there is no under- or overestimation for the whole set of 40 − 176 catchments, but nothing more! After defining the threshold, the interpretation could be different! Besides, it would make sense to normalize over the number of catchments in every regime? And it would be reasonable to cut Y scale for (i) at -50 and +50, even if one box for HBV will not be fully visible.

l. 158: Over all seasons, most models show an acceptable performance (i.e. median error close to zero): → if the median error is close to zero, it does not mean that most models show an acceptable performance!!! It only means that there is no tendency to over- or underestimation for catchments in five regimes, nothing more!

l. 156: Over all, there is no clear tendency of one model to perform better than the

other ones. → Based on thresholds, this could be better visible.

l. 224-225: reliance on an individual calibration metric (EKG) rather than a broader suite of performance metrics can lead to simulation performance deficits for phenomena of interest, including an underestimation of streamflow variability: → Not only the metric, but the calibration approach is general!!!

l. 238: the number of flood events in a simulation time series, which tend to verify well: → disagree, see above!

l. 245-246: Such focus could be improved by giving more weight to the variability component of EKG → or including indicators of extremes in the calibration/validation!!!

Minor corrections needed:

Fig. 1: catchments are indicated by the gauge location?

Fig. 2: for which period(s) is this statistics?

l. 112: circular statistics???

Fig. 3: to explain what is represented by each box with whiskers: comparison for all catchments in a regime over which period: 1981-2008? To add this to the caption.

l. 159-160: particularly in the Western part of the US: → not, in the middle part (intermittent regime)

———

I agree with the authors on the following:

l. 222-223: The results of this study indicate that the limited capability of hydrological models used in this study to reproduce observed hydrologic sensitivities during flooding may be related to insufficient model calibration: FULLY AGREE!

l. 247: The spatial concern could be addressed by applying spatial calibration procedures: → Agree!

l. 254-256: We conclude that calibration using only an individual model performance metric or variable can result in model implementations that have limited value for specific model applications, such as local and in particular spatial flood hazard analyses and change impact assessments: AGREE!

l. 258: more comprehensive multi-objective and multi-variable calibration strategies are needed: AGREE!

References Choi and Beven, 2007, doi:10.1016/j.jhydrol.2006.07.012 Coron et al., 2012, doi:10.1029/2011WR011721 Refsgaard et al., 2013, doi:10.1007/s10584-013-0990-2 Thirel et al., 2015, doi:10.1080/02626667.2015.1050027 Krysanova et al., 2018, DOI: 10.1080/02626667.2018.1446214

---

## Author Comment (AC1) · 23 Jul 2020

**Reviewer 1**

**General comments**

This paper assesses how well hydrological models calibrated using the Kling-Gupta Efficiency (KGE) metric can reproduce local and regional flood characteristics. Stream-flow simulations from four hydrological models are evaluated across a large sample of hydrologically varied catchments, for flood timing, magnitude and spatial variability. In addition, the authors explore the model sensitivities of high flows to precipitation and temperature. This is an interesting analysis and helps to explain model deficiencies for hazard and change impact assessments. I enjoyed reading this paper, which is well-written, concise and easy to follow. The figures are all relevant and well-presented. My main concern is that the title, and focus on deficiencies of integrated calibration metrics, does not accurately reflect the study. I think the title suggests that different model calibration strategies are going to be implemented and evaluated, or that there is going to be some assessment of performance for different calibration strategies. The study only looks at models calibrated using KGE, and it is therefore hard to distinguish if any failure of the models in representing flood characteristics is due to the calibration strategy or other factors such as quality of input and observed streamflow data, or model structural errors. Overall, I think this paper would make an interesting contribution for HESS, following changes and clarifications to the manuscript. I have several specific comments which I outline below.

**Reply:** *Thank you very much for acknowledging the value of our work and for highlighting the need to revise the title. We did indeed not evaluate different calibration strategies. As the title may suggest otherwise, we revised it.*

**Specific comments**

Title: as discussed in general comments, I am not sure the title best reflects the content of the paper. I think this title suggests evaluation of different calibration strategies, whereas KGE has been used throughout. A title focusing on key results/ what has been done (e.g. Evaluating hydrological model suitability for flood impact assessments across a large sample of catchments) may be more suitable.

**Reply:** *Thank you for indicating that the title did not well reflect the content of the manuscript. We replaced the title by: 'Evaluating the suitability of hydrological models for flood impact assessments'.*

Line 10: "Our results show that both the modelling of local and spatial flood characteristics is challenging." It could be helpful to highlight some the key results in the abstract to justify this statement, i.e. all models under predict the magnitude of events.

**Reply:** *We highlight some key results in the abstract such as 'Our results show that both the modeling of local and spatial flood characteristics is challenging as models underestimate flood magnitude and flood timing is not necessarily well captured.'*

Line 12: "We conclude..." The manuscript focuses on models calibrated on KGE alone, and infers that deficiencies in model performance is due to the model calibration. I think this is quite a big leap – as there are other factors which could result in poor model performance (e.g. errors in observed precipitation and river flow data, particularly for peak flow events). It would be good to discuss these within the manuscript.

**Reply:** *Thank you for highlighting that other factors influencing model performance merit more attention. We extended the discussion on input uncertainty (l.195-199) by streamflow observation uncertainty: 'In addition, model performance may depend on the uncertainty of streamflow observations* [McMillan et al., 2010] *used for calibrating and evaluating the model or on input uncertainty, i.e. the precipitation product used to drive the models* [Te Linde et al.,

*2007]). Precipitation products may underestimate extreme rainfall or the spatial dependence of extreme precipitation at different locations because spatial smoothing or averaging during the gridding process reduces variability [Risser et al., 2019].*

Introduction:
Line 25: There is a tendency for high values to be underestimated and low values to be overestimated (Gupta et al. 2009), but I am not sure it is correct to say that the optimal value actually underestimates flow variability. It could be worth mentioning that NSE is often used for high flow studies, based on the idea that by using squared errors it mostly constrains peaks and high flows (Mizukami et al. 2019).
**Reply:** *We think that it is justified to talk about an underestimation of flow variability as an underestimation of high flows and an overestimation of low flows implies an underestimation of flow variability. We mentioned that NSE is widely used for high flow studies and added that using square errors enables focusing on high flows: 'For example, one widely-used metric that is considered integrative compared to others (e.g., bias, correlation) is the Nash-Sutcliffe efficiency (NSE), where the sum-of-sqaures error metric focuses attention on high flow.'*

Line 33: I do not completely follow this sentence – why non-flood-related signature?
**Reply:** *Thank you for pointing out the need for rephrasing. We rephrased the sentence to: 'The use of multiple objectives may, however, lead to a decrease in performance with respect to any individual flow signature not considered as an objective.*

Data and Methods: Whilst the methods section is clear overall, I felt that a few sections needed clarifying.
Line 68: It would be useful to add references for the models.
**Reply:** *We repeat the references provided in the introduction in the methods section.*

Line 68: It would be helpful to know some more about the differences/similarities between the models. In particular, any differences in modelling decisions that may contribute to the performance differences (it would be good to explain why HBV does so poorly compared to the other models). Perhaps a table of key differences or a figure giving model structure diagrams would be helpful.
**Reply:** *Thank you for this suggestion. We have created model structure diagrams to aid in the interpretation of between-model differences, which are provided in the appendix of the manuscript. Reproducing the model equations is infeasible, mainly because of the length of the VIC and mHM code. Instead of reproducing the equations here, we have updated the text with references to where the source code of each model may be found.*

*The model diagrams are based on:*

- *SAC-SMA: analysis of the model's description[National Weather Service NOAA, 2002]: https://www.nws.noaa.gov/oh/hrl/general/chps/Models/Sacramento_Soil_Moisture_Accounting.pdf.*
- *TUW HBV: analysis of the model's source code [Viglione and Parajka, 2020].*
- *VIC: descriptions of VIC in [Melsen et al., 2018; Melsen and Guse, 2019].*
- *mHM: analysis of the model's source code (https://git.ufz.de/mhm/mhm/-/tree/5.7) and a diagram provided in [Kumar et al., 2010].*

*We hypothesize that: 'The overestimation of the number of events by HBV may be explained by its fast response to precipitation as expressed through its model parameter b, which introduces non-linearity to the system [Viglione and Parajka, 2020].'*

Line 70: "model parameters were calibrated on streamflow observations by minimizing the EKG" – How was the optimisation performed (e.g. which algorithm was used) and is this the same in both studies? Was mHM calibrated using multiscale parameter regionalisation, and if so was EKG evaluated across the region rather than for each catchment? It would be useful to know how the calibration differed, despite all being based on KGE.
**Reply:** *We specified that Melsen et al. (2018) used Sobol-based Latin hypercube sampling [Bratley and Fox, 1988] to calibrate VIC, HBV, and SAC. We also specified that mHM was calibrated by Mizukami et al., (2019) for each basin individually using multi-scale parameter regionalization where the transfer function parameters were identified using the dynamically dimensioned search algorithm [Tolson and Shoemaker, 2007].*

Line 80: How do these meteorological forcing data differ? Are they both the same timestep?
**Reply:** *Both forcing datasets are at a daily resolution and both gridded datasets were derived from observed precipitation and temperature. However, the Maurer dataset with 12km has a coarser resolution than the Daymet dataset with 1km. We added these specifications to the text.*

Line 67-83: The dates used for the simulations are unclear. In the method a few different date ranges are given: Line 67: "we use daily streamflow simulations for the period 1981-2008", Line 82: "SAC, HBV and VIC were evaluated on the period 1985-2008", "mHM was calibrated on the period 1999-2008 and evaluated on the period 1989-1999." It seems that 1981-1985 were not used in the previous studies. It would be useful to know which period the model simulations were actually run for, whether a warm-up period has been given, and how long the warmup was. Also, over which period were SAC, HBV and VIC calibrated? Does the period 1981-2008 refer to hydro-logical years or calendar years? It would be helpful to give months here.
**Reply:** *The final analysis was performed on model simulations for the period 1981-2008 for all models. As the model simulations were generated in two different, prior studies, their calibration and evaluation periods do not match as indicated by the different year ranges provided in the text. However, we here used the period 1980-1981 as a spin-up period for all models and performed the analysis on the period 1981-2008. We specified that: 'All four models were finally run for the period 1980-2008 (calendar years), where the period 1980-1981 was used for spin-up and therefore discarded from the analysis.' We also specified that the period 1981-2008 refers to calendar years.*

Line 85: Have you used the KGE values given by Mizukami et al. (2019) and Melsen et al. (2018), or were these re-calculated these over the period 1981-2008? I assumed all model performance was calculated over the same period, against the same observed discharge data, but this is not clear.
**Reply:** *The KGE values were not recalculated over the period 1981-2008, we used the original values provided by Mizukami et al. (2019) and Melsen et al. (2018). We indicate that mHM was evaluated over the period 1989-1999 while the other models were evaluated over the period 1985-2008.*

Line 85: I agree that performance is generally lowest for catchments with intermittent regimes, but there is a lot of overlap in performance.
**Reply:** *We add a note on this stating: 'However, there is a high within-class variability in model performance.'*

Line 114: "we then use the data sets resulting from Step 2 to evaluate how models reproduce overall and seasonal spatial flood dependence." It would be useful to have a bit more detail in this section. How was the error statistic calculated?
**Reply:** *Thank you for pointing out the need for clarification. We specified that: 'We computed actual errors in flood connectedness by subtracting observed from simulated connectedness over all seasons and per season.'*

Line 117: It is not clear if 1% was the value used. This should be made clear, and it would help to have a reference/justification for why this value was chosen.
**Reply:** *We followed the procedure introduced by Brunner et al. (2020) to define flood connectedness. We rephrased the sentence to make this clear: 'Following the definition used by Brunner et al. (2020), a catchment is connected to another catchment if they share a certain number of events. We here used an event threshold of 1% of the total or seasonal number of events to define connectedness (all months: 12 events, seasons: 3 events).*

Line 122: "Time of concentration is typically less than one day for small headwater basins." This needs a reference.
**Reply:** *Besides catchment area, time of concentration also depends on other factors such as rainfall intensity or geology. We therefore made the sentence slightly less specific and provide a reference to a the book chapter by USDA-NRCS (2010).*

Results: A key advantage of this study is the application of multiple model structures to a large sample of catchments. Throughout the methods/results it would be useful to have more discussion of the differences between the models. In particular, it would be useful to know why HBV performs so poorly compared to the other models for flood magnitudes.
Line 145: "For most catchments, the number of flood events is relatively well simulated by most models..." It would be useful to know the number of observed events, to put these errors into context. I am assuming that the number of events is similar between all regime types due to the selection of the threshold. Otherwise a percentage error may be easier to interpret.
**Reply:** *We specify in the Methods section that 'This procedure results in a first quartile of 36, a median of 40, and a third quartile of 47 events identified per basin.' This indicates a relatively small variability in the number of events chosen per basin and justifies the use of actual errors. We provide a model schematic for each of the models considered in this study to aid the interpretation of model differences. We reason that 'The overestimation of the number of events by HBV may be explained by its fast response to precipitation as expressed through its model parameter b, which introduces non-linearity to the system.'*

Line 150: Underestimation of peak flow is attributed to the KGE metric underestimating variability, and spatially lumped model inputs. This could also be due to data errors– for example, McMillan et al. (2012) show that there can be large uncertainties associated with precipitation products. It would be useful to add this to the discussion. McMillan, H., Krueger, T., & Freer, J. (2012). Benchmarking observational uncertain-ties for hydrology: rainfall, river discharge and water quality. Hydrological Processes,26(26), 4078-4111.
**Reply:** *We totally agree that underestimation may also result from data errors. We add the reference to McMillan et al. (2012) to our discussion about the influence of other uncertainty sources than model and parameter choice on flood characteristics (l. 195-199). We specify that: 'Precipitation products may show observation uncertainties [Mcmillan et al., 2012] and underestimate extreme rainfall or the spatial dependence of extreme precipitation at different*

*locations because spatial smoothing or averaging during the gridding process reduces variability* [*Risser et al.*, 2019]*.*

Line 152: "the use of lumped forcings may also artificially synchronize hydrologic response, which would lead to overestimation." – could this be further explained?
**Reply:** *We removed this sentence as it referred to underestimation of spatial dependence rather than magnitude and therefore did not fit into this section.*

Line 163: "the overestimation of spatial dependence in winter is likely related to higher simulated than observed snowmelt." I was not sure which regime(s) this comment was referring to. The melt regime is the only one that doesn't show an overestimation of spatial dependence in winter for any model.
**Reply:** *We specify that: 'The overestimation of spatial dependence in winter for all regimes except the melt regime is likely related to...'*

Line 170: "Connectedness overestimation is most pronounced..." I don't agree with this sentence. For the other 3 models intermittent regime does not seem to be over-estimated any more than other regime types. In winter, it seems to be in line with and below all regimes for SAC, VIC and mHM.
**Reply:** *We agree that this sentence was not correct. We rephrased it to: 'Connectedness overestimation by HBV is most pronounced for catchments with an intermittent regime. Otherwise, connectedness over-/underestimation seems to be independent of the regime.'*

Line 177: "There is a clear positive relationship..." I would not say there is a clear positive relationship for SAC. Perhaps a slight positive relationship.
**Reply:** *We weakened the sentence by deleting the word 'clear'.*

Line 180: "soil moisture and event magnitude are also positively related..." I would interpret this a little differently for VIC - at full saturation we see events of all magnitudes. It is just the upper level of flows that is increasing with soil moisture. 'lower values' -does this mean lower values of peak Q?
**Reply:** *Yes, we specified that we were referring to peak values.*

Line 183: I think this is also the case for SAC.
**Reply:** *Yes, SAC lies somewhere in between. We add the following sentence: 'Such low precipitation inputs can also lead to high peak discharge for SAC but to a lesser degree than HBV and mHM.'*

Line 186: "for SAC and VIC." I would add that to some extent this is also the case for HBV.
**Reply:** *We added: 'and to a lesser degree for HBV'.*

Line 199: Why is this the case?
**Reply:** *Future precipitation estimates are particularly uncertain because of climate model and scenario uncertainty* [*Lopez-Cantu et al.*, 2020]*, which was specified in the text.*

Line 211: "These relationships are, however, not necessarily captured by the models..." It may be worth highlighting that in some areas these relationships are generally captured by the models: e.g. weak winter regime broadly captures the precipitation relationship, and New-Year's regime captures precipitation and temperature relationship.

**Reply:** *We rephrased this section to: 'While these relationships are captured for some catchments (e.g. Blackwater River, weak winter regime or Tucca Creek, New Year's regime), they aren't in other catchments. The simulated sensitivities may even point in another direction than the observed ones (e.g. Pacific Creek, melt regime).'*

Figure 6: This figure has a lot of text, which can be distracting from the plots. I think it would be help to simplify the y axes and colorbar scales to 2 significant figures (i.e. no decimal places).
**Reply:** *We agree that Figure 6 would profit from 'decluttering'. We reduced the number of digits displayed to 1 wherever possible and added colored boxes to improve the reading flow.*

Figure 6: It would be clearer if the colour scales matched between the observations and simulations for a specific catchment, and also the x and y axis ranges. Otherwise, it would be useful to point out that the scales differ, and explain why this has been done, i.e. colours ranging from the largest to smallest flood.
**Reply:** *We chose different color scales and axes for the observed and simulated floods to compare gradients rather than differences in magnitudes. Plotting the grids on the same grid and using the same colors would lead to non-centered grid clouds and to very weak colors in the case floods are underestimated. We adjusted the figure caption and point out that the different grids are shown on different scales: 'Grid axes and grey scales differ between plots where darker colors indicate higher flood magnitudes.'*

Line 221: I do not follow this link -could this be explained better? It feels like there is a jump from models inadequately representing the sensitivity of peak flows to precipitation to errors in precipitation data being the cause.
**Reply:** *We agree that the link between the two sub-sentences was not evident. As we discuss precipitation errors as a potential source elsewhere in the manuscript (l.195-199), we removed its mention from here.*

Line 223: ".. may be related to insufficient model calibration..." This feels like quite a big leap. Having only looked at models calibrated using KGE it doesn't feel like there is enough information to attribute poor performance to calibration metrics. Could it be the model structures more generally, or the input data errors, that are causing these model deficiencies rather than the calibration metric?
**Reply:** *Yes, model structure and input data uncertainty are definitely also part of the story. We acknowledge this in the newly phrased paragraph: 'The results of this study indicate that the hydrological models used in this study have limited capability in reproducing observed hydrologic sensitivities during flooding. These limitations may be related to input uncertainties [Te Linde et al., 2007], equifinality in process contributions for simulations with (very) similar efficiency scores, leading to an inability to unambiguously identify the appropriate relative process contributions [Khatami et al., 2019] or insufficient model calibration [Fowler et al., 2016].'*

Figure 7: It would be helpful to have a more thorough explanation of this figure. Perhaps a sentence explaining that positive values mean an increase in the variable leads to an increase in peak flows, and values falling on the dotted line indicate simulations match observations.
**Reply:** *Thank you for pointing out the need for clarification. We added that: 'Positive and negative values indicate positive and negative associations of precipitation and temperature with peak flow, respectively. Values on the dashed line indicate correspondence between observed and modeled sensitivity gradients.'*

Line 226: This sentence implies an underestimation in timing. Only absolute errors in day of flood timing are given, not the direction of change within the year. Maybe rephrase this sentence.
**Reply:** *We rephrased this to: 'including an underestimation of streamflow variability and peak flood magnitudes and a misrepresentation of timing.'*

Conclusions:
Line 235: In the introduction a key aim is 'assess which aspects of hydrological models may need to be improved ....' and 'identifying and documenting model weaknesses regarding regional and future flooding will highlight advances for future model development.' .. These aims/questions could be more directly addressed in the conclusions section.
**Reply:** *We try to more explicitly address these aims by adding: 'We therefore conclude that the representation of magnitude, timing and spatial connectedness can be improved.'*

**Technical corrections**
Line 86: "successfully" should be "success"
**Reply:** *We think that the phrasing is correct and retained successfully.*

Line 160: "underestimates" should be "underestimate"
**Reply:** *We removed the 's'.*

**HESS Review Checklist**
In the full review and interactive discussion, the referees and other interested members of the scientific community are asked to take into account all of the following aspects:
1) Does the paper address relevant scientific questions within the scope of HESS? YES
2) Does the paper present novel concepts, ideas, tools, or data? YES
3) Are substantial conclusions reached? YES
4) Are the scientific methods and assumptions valid and clearly outlined? YES
5) Are the results sufficient to support the interpretations and conclusions? MOSTLY
6) Is the description of experiments and calculations sufficiently complete and precise to allow their reproduction by fellow scientists (traceability of results)? YES
7) Do the authors give proper credit to related work and clearly indicate their own new/original contribution? YES
8) Does the title clearly reflect the contents of the paper? NO
9) Does the abstract provide a concise and complete summary? YES
10) Is the overall presentation well structured and clear? YES
11) Is the language fluent and precise?
YES12) Are mathematical formulae, symbols, abbreviations, and units correctly defined and used?
YES13) Should any parts of the paper (text, formulae, figures, tables) be clarified, reduced,combined, or eliminated? MINOR CLARIFICATIONS TO METHODS
14) Are the number and quality of references appropriate? YES
15) Is the amount and quality of supplementary material appropriate? YES
**Reply:** *We changed the title and added a few clarifications to the methods section as discussed in detail in the individual comments above.*

**References used in this response to the reviewer**

Bratley, P., and B. L. Fox (1988), Algorithm 659: Implementing Sobol's Quasirandom Sequence Generator, *ACM Trans. Math. Softw.*, *14*(1), 88–100, doi:10.1145/42288.214372.

Brunner, M. I., E. Gilleland, A. Wood, D. L. Swain, and M. Clark (2020), Spatial dependence of floods shaped by spatiotemporal variations in meteorological and land-surface processes, *Geophys. Res. Lett.*, *47*, e2020GL088000, doi:10.1029/2020GL088000.

Fowler, K. J. A., M. C. Peel, A. W. Western, L. Zhang, and T. J. Peterson (2016), Simulating runoff under changing climatic conditions: Revisiting an apparent deficiency of conceptual rainfall-runoff models, *Water Resour. Res.*, *52*, 1820–1846, doi:10.1111/j.1752-1688.1969.tb04897.x.

Khatami, S., M. C. Peel, T. J. Peterson, and A. W. Western (2019), Equifinality and flux mapping: A new approach to model evaluation and process representation under uncertainty, *Water Resour. Res.*, *55*(11), 8922–8941, doi:10.1029/2018WR023750.

Kumar, R., L. Samaniego, and S. Attinger (2010), The effects of spatial discretization and model parameterization on the prediction of extreme runoff characteristics, *J. Hydrol.*, *392*(1–2), 54–69, doi:10.1016/j.jhydrol.2010.07.047.

Te Linde, A. H., J. Aerts, H. Dolman, and R. Hurkmans (2007), Comparing model performance of the HBV and VIC models in the Rhine basin, in *Quantification and Reduction of Predictive Uncertainty for Sustainable Water Resources Management*, pp. 278–285.

Lopez-Cantu, T., A. F. Prein, and C. Samaras (2020), Uncertainties in future U.S. extreme precipitation from downscaled climate projections, *Geophys. Res. Lett.*, *47*(9), 1–11, doi:10.1029/2019GL086797.

Mcmillan, H., T. Krueger, and J. Freer (2012), Benchmarking observational uncertainties for hydrology: Rainfall, river discharge and water quality, *Hydrol. Process.*, *26*(26), 4078–4111, doi:10.1002/hyp.9384.

McMillan, H., J. Freer, F. Pappenberger, T. Krueger, and M. Clark (2010), Impacts of uncertain river flow data on rainfall-runoff model calibration and discharge predictions, *Hydrol. Process.*, *24*(10), 1270–1284, doi:10.1002/hyp.7587.

Melsen, L., N. Addor, N. Mizukami, A. Newman, P. Torfs, M. Clark, R. Uijlenhoet, and R. Teuling (2018), Mapping (dis) agreement in hydrologic projections, *Hydrol. Earth Syst. Sci.*, *22*, 1775–1791, doi:10.5194/hess-22-1775-2018.

Melsen, L. A., and B. Guse (2019), Hydrological drought simulations: How climate and model structure control parameter sensitivity, *Water Resour. Res.*, 1–21, doi:10.1029/2019wr025230.

Mizukami, N., O. Rakovec, A. J. Newman, M. P. Clark, A. W. Wood, H. V. Gupta, and R. Kumar (2019), On the choice of calibration metrics for "high-flow" estimation using hydrologic models, *Hydrol. Earth Syst. Sci.*, *23*(6), 2601–2614, doi:10.5194/hess-23-2601-2019.

National Weather Service NOAA (2002), *Conceptualization of the Sacramento Soil Moisture accounting model*.

Risser, M. D., C. J. Paciorek, M. F. Wehner, T. A. O'Brien, and W. D. Collins (2019), A probabilistic gridded product for daily precipitation extremes over the United States, *Clim. Dyn.*, *53*(5–6), 2517–2538, doi:10.1007/s00382-019-04636-0.

Tolson, B. A., and C. A. Shoemaker (2007), Dynamically dimensioned search algorithm for computationally efficient watershed model calibration, *Water Resour. Res.*, *43*(1), 1–16, doi:10.1029/2005WR004723.

USDA-NRCS (2010), Time of concentration, in *National Engineering Handbook: Part 630 Hydrology*, pp. 1–15, U.S. Department of Atriculture (USDA), Fort Worth.

Viglione, A., and J. Parajka (2020), TUWmodel: Lumped/Semi-Distributed Hydrological Model for

Education Purposes, *TUWmodel Lumped/Semi-Distributed Hydrol. Model Educ. Purp.*, https://cran.r-project.org/web/packages/TUWmodel/i.  Available from: https://cran.r-project.org/web/packages/TUWmodel/index.html (Accessed 25 June 2020)

---

## Author Comment (AC2) · 23 Jul 2020

**Reviewer 2**

This is a well-written journal with appropriate content for HESS. I think this is a nice study, though I do have some suggestions related to the framing of the work and its discussion. This could be a very nice paper if the focus was actually on the calibration strategy.

**My comments are:**
[1] The title of the study suggests a wide-ranging assessment of different calibration strategies in the context of flood modelling. However, the study is essentially an assessment of the value of using KGE for flood modelling. The actual focus is fine, but I think it should be reflected in the title of the manuscript to avoid confusion.
**Reply:** *Thank you for pointing out the mismatch between the title and the analyses performed. This point was also raised by reviewer 1 and we changed the title to: 'Evaluating the suitability of hydrological models for flood impact assessments.' This rephrasing removes the emphasis from model calibration, whose effect on model simulations has been assessed by Mizukami et al. (2019).*

[2] Given that the focus of the manuscript is on the calibration strategy, I was surprised to not find any details on what strategy was used to find the best KGE values? What algorithm was used etc would be helpful information for the reader to understand what has been done. While this might be covered in previous papers in detail, it would be good to see at least some basic description here as well.
**Reply:** *We specified that: 'The model parameters were calibrated on streamflow observations by minimizing the $1-E_{KG}$ by Melsen et al. (2018) using Sobol-based Latin hypercube sampling [Bratley and Fox, 1988] for SAC, HBV, and VIC and by Mizukami et al. (2019) for mHM using multi-scale parameter regionalization where the transfer function parameters were identified using the dynamically dimensioned search algorithm [Tolson and Shoemaker, 2007].'*

[3] It would also be helpful to have some calibration/validation results for each model to distinguish them at this point already (if they differ?).
**Reply:** *We provide validation results for each of the four models in Figure 2 and specify that: 'Overall model performance decreases from mHM (median $E_{KG}$ 0.69), over SAC (median $E_{KG}$ 0.63) and VIC (median $E_{KG}$ 0.60) to HBV (median $E_{KG}$ 0.52).' So yes, the models are already different if we just look at $E_{KG}$ before considering any specific flood metric.*

[4] Section 3.1: Why is HBV so poor? Especially given its focus on snow/cold regions?
**Reply:** *It is difficult to say why exactly HBV is performing worse than the other three models in reproducing flood characteristics. We think that: 'The overestimation of the number of events by HBV may be explained by its fast response to precipitation as expressed through its model parameter b, which introduced non-linearity to the system [Viglione and Parajka, 2020].' and added this statement to the text.*

[5] Section 3.1: I am a bit confused by this assessment. Are you assessing the model or the metric used for calibration? The paper title suggests that the focus is on the calibration strategy, so my question is why using the same calibration strategy results in different model performance? Significant differences between very similar models is surprising if the models have been calibrated in the same manner.
**Reply:** *We agree that the choice of the initial title could cause some confusion. Instead of comparing different calibration strategies as done in previous studies [Mizukami et al., 2019], we compare the representation of floods by different models calibrated with the same objective*

*function. As mentioned above, we have changed the title in order to eliminate focus on the calibration strategy itself. Our results show that even if models are calibrated using a calibration metric supposedly putting a lot of weight on high flows, they may not necessarily well represent local and regional features of floods.*

[6] Lines 224-225: But how do you know that if you only assessed one metric? The authors do a very nice job of including multiple models, but if the focus is on the calibration strategy, then why do you not include variability in how they calibrate the models? How can you make conclusions about the calibration strategy if you did not vary it. Would putting more weight on fitting the variability have produced a better fit to variability (using a weighted KGE)? You have this as a discussion point, but why is this not part of your actual study?
**Reply:** *As wrongly suggested by our initial title, the focus of this study is not supposed to be on the calibration strategy as the effect of the choice of an objective function on the quality of modeled flood flows has previously been assessed by Mizukami et al. (2019). They show that $E_{KG}$ leads to a better model performance with respect to flood flows than $E_{NS}$, which is very often recommended for calibrating a model aimed at simulating flood peaks/high flows. We show that even if one uses the metric found to lead to the best flood representation by Mizukami et al. (2019), the reproduction of flood characteristics may still leave much to be desired. We rephrased this sentence to: 'We illustrate that reliance on an individual calibration metric ($E_{KG}$) can lead to simulation performance deficits for phenomena of interest, including an underestimation of streamflow variability and peak flood magnitudes and a misrepresentation of timing'*

[7] Line 236: But how do you know that? Maybe all the models have the same problem regardless of calibration metric used? Maybe you did not look hard enough for an optimum parameter set?
**Reply:** *Our results show that models do not perform equally well in simulating flood characteristics when calibrated with the same objective function. We therefore think that the statement 'Our model comparison shows that all flood characteristics are not equally well represented by models calibrated with the widely used Kling–Gupta efficiency metric' is justified. We acknowledge that these limitations may not solely be related to model structure: 'These limitations may be related to input uncertainties [Te Linde et al., 2007], equifinality in process contributions for simulations with (very) similar efficiency scores, leading to an inability to unambiguously identify the appropriate relative process contributions [Khatami et al., 2019] or insufficient model calibration [Fowler et al., 2016].*

[8] Line 245: As stated above, I find it dissatisfying to make such a conclusion. Testing this suggestion is a very minor effort given the work already presented in this paper.
Why can the authors not try this? This – to me – would be part of the main tests the authors should have done in this paper. You cannot test the implications of choices about the calibration strategy if you do not test different choices. Using multiple models does not compensate for this omission.
**Reply:** *As discussed above, the focus of this study was not supposed to be on a comparison of different model calibration strategies even though our initial title may have suggested otherwise. Rather, we wanted to show that using a calibration metric commonly recommended for model calibration in the case one is interested in floods may still lead to suboptimal model results. The development of an objective function targeted at optimizing local and spatial flood characteristics would be a study in itself. This is why we leave potential ways of improving calibration strategies to the discussion.*

**References used in this response to the reviewer**

Bratley, P., and B. L. Fox (1988), Algorithm 659: Implementing Sobol's Quasirandom Sequence Generator, *ACM Trans. Math. Softw.*, *14*(1), 88–100, doi:10.1145/42288.214372.

Fowler, K. J. A., M. C. Peel, A. W. Western, L. Zhang, and T. J. Peterson (2016), Simulating runoff under changing climatic conditions: Revisiting an apparent deficiency of conceptual rainfall-runoff models, *Water Resour. Res.*, *52*, 1820–1846, doi:10.1111/j.1752-1688.1969.tb04897.x.

Khatami, S., M. C. Peel, T. J. Peterson, and A. W. Western (2019), Equifinality and flux mapping: A new approach to model evaluation and process representation under uncertainty, *Water Resour. Res.*, *55*(11), 8922–8941, doi:10.1029/2018WR023750.

Te Linde, A. H., J. Aerts, H. Dolman, and R. Hurkmans (2007), Comparing model performance of the HBV and VIC models in the Rhine basin, in *Quantification and Reduction of Predictive Uncertainty for Sustainable Water Resources Management*, pp. 278–285.

Melsen, L., N. Addor, N. Mizukami, A. Newman, P. Torfs, M. Clark, R. Uijlenhoet, and R. Teuling (2018), Mapping (dis) agreement in hydrologic projections, *Hydrol. Earth Syst. Sci.*, *22*, 1775–1791, doi:10.5194/hess-22-1775-2018.

Mizukami, N., O. Rakovec, A. J. Newman, M. P. Clark, A. W. Wood, H. V. Gupta, and R. Kumar (2019), On the choice of calibration metrics for "high-flow" estimation using hydrologic models, *Hydrol. Earth Syst. Sci.*, *23*(6), 2601–2614, doi:10.5194/hess-23-2601-2019.

Tolson, B. A., and C. A. Shoemaker (2007), Dynamically dimensioned search algorithm for computationally efficient watershed model calibration, *Water Resour. Res.*, *43*(1), 1–16, doi:10.1029/2005WR004723.

Viglione, A., and J. Parajka (2020), TUWmodel: Lumped/Semi-Distributed Hydrological Model for Education Purposes, *TUWmodel Lumped/Semi-Distributed Hydrol. Model Educ. Purp.*, https://cran.r-project.org/web/packages/TUWmodel/i. Available from: https://cran.r-project.org/web/packages/TUWmodel/index.html (Accessed 25 June 2020)

---

## Author Comment (AC3) · 23 Jul 2020

**Reviewer 3**

Though I agree with the title of the manuscript and with the main conclusions on l.254-259 (see below), there are several serious deficiencies in the approach and interpretation of results. The study looks like an initial stage only: let us calibrate four models for many relatively small catchments in USA using one metric, EKG, to see, how well the models will reproduce local flood characteristics and spatial aspects of flooding, and how well would they be prepared for climate impact assessment. The conclusion is that the models calibrated on the Kling–Gupta efficiency alone have limited reliability in flood hazard assessments. Such "negative" result could be expected, as there are several recent publications pointing on a necessity of comprehensive approaches for hydrological model calibration and evaluation (for mean flow and for extremes) and especially if they are intended for climate impact assessment (see e.g. Choi and Beven, 2007, Coron et al.,2012, Refsgaard et al., 2013, Thirel et al., 2015, Krysanova et al., 2018). Therefore, such an "initial stage" of the study should be supplemented by application of an extended approach: for example, including at least some of the further steps suggested in the papers listed above, like multi-site and multi-variable calibration (mentioned in the manuscript), DSS test checking for contrasting climate sub-periods, testing specifically for indicators of interest, i.e. for high flows and floods. Then the study would be much more valuable. There are also other deficiencies in the applied approach and in the interpretation of the obtained results. Therefore, the manuscript should be rejected in its present form.

**Reply:** *Thank you for your time reviewing our manuscript. We agree that the results highlight model deficiencies in the representation of local and regional flood characteristics particularly under climate conditions different from the current ones. We also agree that some previous studies have tried to highlight the necessity to evaluate model transferability to conditions different from the ones we live in. Still, we are surprised by how many studies (including some of our own studies) use $E_{NS}$ or $E_{KG}$ as a single calibration and evaluation metric for flood studies (for $E_{NS}$ based calibration see e.g. [Hundecha and Merz, 2012; Köplin et al., 2014; Vormoor et al., 2015; Wobus et al., 2017] and for $E_{KG}$ based calibration see e.g. [Brunner and Sikorska, 2018; Hirpa et al., 2018; Huang et al., 2018; Thober et al., 2018; Harrigan et al., 2020]. The aim of this study is to clearly communicate to the hydrologic modeling community that such a focus on a single metric may not result in an accurate representation of flood characteristics, particularly not in a spatial and climate change context. Our study should contribute to expanding awareness of such issues within a field that we observe continues to rely (too) strongly on the $E_{KG}$ and like metrics alone. We focused on $E_{KG}$ because a previous study by [Mizukami et al., 2019] has shown that $E_{KG}$ results in a more reliable representation of peak discharge than $E_{NS}$.*

*In all, we feel the presentation of multiple analyses of different aspects of model behavior for four models and hundreds of locations to shed further light on aspects of the simulations that are present but not directly indicated from a single $E_{KG}$ score (which may be high) goes beyond an initial analysis. We do pay particular attention to the representation of spatial flood characteristics and the suitability of model setups in simulating floods under climate conditions different from the ones in the observations. To do so, we perform a resampling-based sensitivity analysis focusing on peak-over-threshold values, which has similar aims as the differential split sample test. We try to better explain this similarity by adding the following description to the introduction of this methodology: 'To do so, we look at how models translate changes in event temperature and precipitation into changes in POT discharge by performing a resampling-based sensitivity analysis. This sensitivity analysis aims at evaluating whether a model is still reliable under climate conditions different from the ones used in model calibration similar to split-sample or differential split-sample calibration/validation schemes [Coron et al., 2012; Refsgaard et al., 2014; Thirel et al., 2015].'*

*Besides highlighting potential modeling challenges related to the representation of floods, our discussion section points out potential avenues for further model improvement including the use of more tailored model calibration strategies, giving more weight to the variability component of an integrative metric, optimizing explicitly for key flood characteristics (e.g., peak flow, volume, timing) and/or metrics depicting the fidelity of the model representation of soil moisture and snowmelt, using a multi-objective model calibration process, or considering spatially distributed features of model response within a spatial calibration framework. This is thought as an encouragement to researchers who work on the development of innovative calibration techniques rather than an attempt to propose actual alternative calibration metrics ourselves. While we share the view that an expanded study which goes on to test the hypothesis that multi-objective, signature-aware and other types of calibration approaches would lead to more suitable models for flooding and change studies, adding that evidence to this study is not feasible given the substantial effort and time involved.*

**Other major concerns:**

APPROACH

l. 81-82: were driven with Daymet meteorological forcing (Thornton et al., 2012) and mHM with the forcing by Maurer et al. (2002):→how are they comparable with the observed climate? Was the comparison done or not? If not, it would be reasonable to do.

**Reply:** *Thank you for pointing out the need for clarification. Both the Daymet and Maurer datasets represent current climate conditions, were derived from observed precipitation and temperature, and have been shown to result in similar mean daily precipitation fields [Newman et al., 2015]. We specified that 'All the models were driven with daily, spatially lumped meteorological forcing data representing current climate conditions: SAC, HBV, and VIC were driven with Daymet meteorological forcing (1km resolution) and mHM with the forcing by Maurer et al. (2002) (12km resolution) both derived from observed precipitation and temperature. So yes, they represent observed climate.*

l. 82: SAC, HBV, and VIC were evaluated on the period 1985–2008:→and calibrated for which period?

**Reply:** *Melsen et al. (2018) ran the three models using a large number of parameter sets for the period 1985-2008. These parameter sets were generated by first sampling 100 base runs based on the average parameter values. Subsequently, they sampled each parameter 100 times by applying perturbations to the base runs. This implies that for each of the 605 basins, SAC was run 1900 times, VIC 1800 times, and HBV 1600 times. From these runs, we here chose the best parameter set in terms of $E_{KG}$, which represents the calibration step in a wider sense as the definition of calibration is identifying parameters. This procedure does not correspond to a classical calibration-validation scheme where the model is evaluated over a validation period independent of the calibration period but rather to a sampling procedure.*

l. 110-112:→would be good to express the relative error in %, and define thresholds for acceptable performance (e.g. based on literature) for all 3 indicators. For example, is a relative error of 25% acceptable or not? The thresholds could be shown in Fig. 3 by horizontal lines to enable distinguishing the good/acceptable and poor performances. Sec. 3.1:→to discuss performance based on the pre-defined thresholds

**Reply:** *We expressed the relative errors in Figure 3 in %. Defining a threshold for acceptable performance is a great idea. However, such an acceptability threshold is likely to depend on the problem at hand and a general threshold is therefore difficult to define.*

l. 116-118: a catchment is connected to another catchment if they share a certain number of events, i.e. at least 1% of the total or seasonal number of events:→is 1%of shared events really sufficient to define their connectivity??? Due to that, the whole section 3.2 is questionable.

**Reply:** *Thank you for expressing your concern and highlighting the need for clarification. We provide additional information on how many flood events were in the data set and how this translates into thresholds used: 'To do so, we use the connectedness measure introduced by Brunner et al. (2020), which quantifies the number of catchments with which a specific catchment co-experiences floods. The number of concurrent flood events for a pair of stations is determined based on a data set consisting of the dates of flood occurrences across all catchments. This set is converted into a binary matrix which specifies for each catchment whether or not it is affected by a certain event. The matrix compiled using observed streamflow time series contained 1164 events among which 258 occur in winter, 291 in spring, 324 in summer, and 291 in fall. Following the definition used by Brunner et al. (2020), a catchment is connected to another catchment if they share a certain number of events. We here used an event threshold of 1% of the total or seasonal number of events to define connectedness (all months: 12 events, seasons: 3 events).' These values are similar to the absolute values used by Brunner et al. (2020) who used thresholds of 10 events for the annual and 5 events for the seasonal analysis. We consider these thresholds high enough to avoid defining a pair of stations as connected coincidentally.*

l. 127: we generate surrogate time series of temperature, precipitation, and streamflow for each catchment by resampling the available hydrological years with replacement:→the procedure is not quite clear, and should be better explained!

**Reply:** *Thank you for pointing out the need to provide more specifics on the sampling strategy. We specify that: 'To generate these series, we randomly sample a series of years with replacement in the period 1981-2008 which we use to compose time series consisting of the daily values corresponding to these years for each of the three variables'*

l. 202-203: "to assess each model' suitability for climate impact assessments on floods":→how the resampling could help to assess suitability? It would be better to test for contrasting climate sub periods, or to compare trends in discharge, high flows and POT series.

**Reply:** *This resampling procedure allows us to look at whether the models react to changes in mean event temperature and precipitation in the same way as the real world system. E.g. if higher observed event precipitation results in higher observed peak discharge, this should ideally be reflected in the modeling system which should show higher peak discharge for events with higher precipitation (i.e. the gradients derived from the observed and simulated response surfaces should be similar). If the model does not reproduce this behavior, its process representation in terms of floods is probably not ideal. We agree that differential split sample testing would be another way of looking at how transferable a model is to climate conditions, which differ from the ones used for calibrating and validating the model [Seibert, 2003]. However, we think that our resampling procedure goes beyond a split sample test because it enables analyzing gradients in the P-T-Q space instead of just comparing two periods that might differ with respect to certain characteristics. Within the observation period (1981-2008) less than 5% of the 488 catchments show statistically significant trends in POT values according to the non-parametric Mann-Kendall test. A comparison of observed vs. simulated trends is therefore not going to be a very useful evaluation metric with respect to the transferability of the model to changed climate conditions.*

Sec. 3.3 and Fig.5:→Maybe to add correlation coefficients to better characterize the relationships?

**Reply:** *Thank you for this suggestion. We added Kendall's rank correlation coefficients to each of the subplots to characterize the relationships between the pair of variables.*

INTERPRETATION

l. 145-146: "For most catchments, the number of flood events is relatively well simulated by most models": →this is not evident, if a threshold is not defined. It is only visible that medians are close to zero for three models, and there is no under- or over-estimation for the whole set of 40 – 176 catchments, but nothing more! After defining the threshold, the interpretation could be different! Besides, it would make sense to normalize over the number of catchments in every regime? And it would be reasonable to cut Y scale for (i) at -50 and +50, even if one box for HBV will not be fully visible.

**Reply:** *Thank you for these suggestions. We scaled the axis of panel (i) to -50 and +50. We indicate the number of catchments per regime in the figure caption to highlight that not all regimes have the same sample size. However, we do not understand your suggestion to normalize as each catchment represents one data point forming the boxplot. We agree that a threshold of model acceptability would be desirable and think that such a threshold would depend on the problem at hand. It is therefore difficult to define a generally valid threshold separating bad from good model performance. To not make any specific judgement, we rephrase the sentence and specify the actual error ranges for each of the models rather than talking about good and bad model performance in the updated version of the manuscript: 'For most catchments, the median deviation between the simulated and observed number of flood events lies close to zero (SAC: -3 events, HBV: -1, VIC: -1, mHM: 0). However, the simulations result in over- and underestimations of the number of events depending on the catchment (1st and 3rd quartiles for SAC: -9, 4; HBV: -8, 15; VIC: -7, 6; mHM: -6, 6). The overestimation is strongest for HBV, which overestimates the number of events for catchments with intermittent, weak winter, and melt regimes.' To still provide some guidance for the reader, we included different thresholds in Figure 2. For each model and regime, we broke up the results into three categories (and boxplots): all catchments, catchments with $E_{KG} > 0.5$, and catchments with $E_{KG} > 0.7$. For the last two categories, we provide the percentage of catchments in the regime under consideration exceeding the respective threshold.*

l. 158: Over all seasons, most models show an acceptable performance (i.e. median error close to zero): →if the median error is close to zero, it does not mean that most models show an acceptable performance!!! It only means that there is no tendency to over- or underestimation for catchments in five regimes, nothing more!

**Reply:** *Good point. We agree that a median error of zero does not necessarily imply acceptable model performance as positive and negative errors can cancel out. We rephrase this sentence in a neutral tone: 'Over all seasons, most models show a median error close to zero for flood connectedness. Flood connectedness can be over- and underestimated dependent on the catchment by most of the models while HBV overestimates spatial dependence in most catchments particularly in the Western part of the US.'*

l. 156: Over all, there is no clear tendency of one model to perform better than the other ones. →Based on thresholds, this could be better visible.

**Reply:** *This might be true with respect to a specific application, where one is interested in a specific regime or flood characteristic. We here intended to make a statement valid independent of a problem and therefore refrained from setting a 'somewhat arbitrary' threshold for model performance. As shown in Figures 3 and 4, SAC, VIC, and mHM perform similarly well regarding most of the flood characteristics assessed here and we therefore think that this statement is valid. We add that: 'However, there are slight differences in model performance which suggests*

*that a 'most suitable model' could be identified for a specific application at hand, where a certain region or variable is of interest.'*

l. 224-225: reliance on an individual calibration metric (EKG) rather than a broader suite of performance metrics can lead to simulation performance deficits for phenomena of interest, including an underestimation of streamflow variability:→Not only the metric, but the calibration approach is general!!!
**Reply:** *Yes, we agree with the reviewer, that the calibration approach/strategy, which includes the selection of time periods for training and validating model skill, screening for sensitive model parameters, and selecting a number of model evaluation metrics, includes a large number of subjective choices and might influence model results. We therefore based this manuscript on previously published work and we believe best possible calibration settings given past computer and resources availability. By changing the title and removing the work "calibration" from it, we believe to have removed the focus on any new calibration exercise/strategy.*

l. 238: the number of flood events in a simulation time series, which tend to verify well:→disagree, see above!
**Reply:** *We agree that a more nuanced statement is required here and rephrased the sentence to: 'The number of floods, flood magnitude, and timing are not always well captured by hydrological models in many catchments. The number of flood events were over- or underestimated depending on the catchment, flood magnitudes were underestimated by all models in most catchments, and the ability of the model to accurately reproduce event timing was proportional to the hydroclimatic seasonality.'*

l. 245-246: Such focus could be improved by giving more weight to the variability component of EKG →or including indicators of extremes in the calibration/validation!!!
**Reply:** *Yes, we tried to express this by writing: or by using a suite of appropriate and targeted metrics in a multi-objective framework. We rephrased this to: 'or by including indicators of extremes in in a multi-objective framework when calibrating and validating the model.'*

**Minor corrections needed:**
Fig. 1: catchments are indicated by the gauge location?
**Reply:** *Correct. We clarify this in the figure caption: 'Map of the 488 catchments in the conterminous United States belonging to the five regime classes indicated by their gauge location.'*

Fig. 2: for which period(s) is this statistics?
**Reply:** *These values refer to the period 1981-2008, which was specified in the figure caption.*

l. 112: circular statistics???
**Reply:** *We specify that: 'circular statistics are suitable for defining central tendencies of variables with a cycle [*Burn*, 1997].*

Fig. 3: to explain what is represented by each box with whiskers: comparison for all catchments in a regime over which period: 1981-2008? To add this to the caption.
**Reply:** *We clarify in the figure caption that: 'The errors were computed over the period 1981-2008', that we looked at 'mean' errors for magnitude and timing and that 'The boxplots are composed of one value per catchment belonging to the respective regime class.'*

l. 159-160: particularly in the Western part of the US:→not, in the middle part (intermittent regime)

**Reply:** *The Western part is actually correct. Because we do not explicitly show this, we removed this sub-sentence though.*

**I agree with the authors on the following:**
l. 222-223: The results of this study indicate that the limited capability of hydrological models used in this study to reproduce observed hydrologic sensitivities during flooding may be related to insufficient model calibration: FULLY AGREE!
l. 247: The spatial concern could be addressed by applying spatial calibration procedures:→Agree!
l. 254-256: We conclude that calibration using only an individual model performance metric or variable can result in model implementations that have limited value for specific model applications, such as local and in particular spatial flood hazard analyses and change impact assessments: AGREE!
l. 258: more comprehensive multi-objective and multi-variable calibration strategies are needed: AGREE!

**Reply:** *We are glad that we have some common ground here.*

**References**
Choi and Beven, 2007, doi:10.1016/j.jhydrol.2006.07.012
Coron et al.,2012, doi:10.1029/2011WR011721
Refsgaard et al., 2013, doi:10.1007/s10584-013-0990-2
Thirel et al., 2015, doi:10.1080/02626667.2015.1050027
Krysanova et al.,2018, DOI: 10.1080/02626667.2018.1446214

**References used in this response to the reviewers**

Brunner, M. I., and A. E. Sikorska (2018), Dependence of flood peaks and volumes in modeled runoff time series: effect of data disaggregation and distribution, *J. Hydrol.*, *572*, 620–629, doi:10.1016/j.jhydrol.2019.03.024.

Brunner, M. I., E. Gilleland, A. Wood, D. L. Swain, and M. Clark (2020), Spatial dependence of floods shaped by spatiotemporal variations in meteorological and land-surface processes, *Geophys. Res. Lett.*, *47*, e2020GL088000, doi:10.1029/2020GL088000.

Burn, D. H. (1997), Catchment similarity for regional flood frequency analysis using seasonality measures, *J. Hydrol.*, *202*, 212–230.

Coron, L., V. Andréassian, C. Perrin, J. Lerat, J. Vaze, M. Bourqui, and F. Hendrickx (2012), Crash testing hydrological models in contrasted climate conditions: An experiment on 216 Australian catchments, *Water Resour. Res.*, *48*(5), 1–17, doi:10.1029/2011WR011721.

Harrigan, S., E. Zsoter, L. Alfieri, C. Prudhomme, P. Salamon, C. Barnard, H. Cloke, and F. Pappenberger (2020), GloFAS-ERA4 operational global river discharge reanalysis 1979-present, *Earth Syst. Sci. Data*, (January), 1–23.

Hirpa, F. A., P. Salamon, H. E. Beck, V. Lorini, L. Alfieri, E. Zsoter, and S. J. Dadson (2018), Calibration of the Global Flood Awareness System (GloFAS) using daily streamflow data, *J. Hydrol.*, *566*(September), 595–606, doi:10.1016/j.jhydrol.2018.09.052.

Huang, S., R. Kumar, O. Rakovec, V. Aich, X. Wang, L. Samaniego, S. Liersch, and V. Krysanova (2018), Multimodel assessment of flood characteristics in four large river basins at global warming of

1.5, 2.0 and 3.0 K above the pre-industrial level, *Environ. Res. Lett.*, *13*(12), 124005, doi:10.1088/1748-9326/aae94b.

Hundecha, Y., and B. Merz (2012), Exploring the relationship between changes in climate and floods using a model-based analysis, *Water Resour. Res.*, *48*(4), doi:10.1029/2011WR010527.

Köplin, N., B. Schädler, D. Viviroli, and R. Weingartner (2014), Seasonality and magnitude of floods in Switzerland under future climate change, *Hydrol. Process.*, *28*(4), 2567–2578, doi:10.1002/hyp.9757.

Melsen, L., N. Addor, N. Mizukami, A. Newman, P. Torfs, M. Clark, R. Uijlenhoet, and R. Teuling (2018), Mapping (dis) agreement in hydrologic projections, *Hydrol. Earth Syst. Sci.*, *22*, 1775–1791, doi:10.5194/hess-22-1775-2018.

Mizukami, N., O. Rakovec, A. J. Newman, M. P. Clark, A. W. Wood, H. V. Gupta, and R. Kumar (2019), On the choice of calibration metrics for "high-flow" estimation using hydrologic models, *Hydrol. Earth Syst. Sci.*, *23*(6), 2601–2614, doi:10.5194/hess-23-2601-2019.

Newman, A. J., M. P. Clark, J. Craig, B. Nijssen, A. Wood, E. Gutmann, N. Mizukami, L. Brekke, and J. R. Arnold (2015), Gridded Ensemble Precipitation and Temperature Estimates for the Contiguous United States, *J. Hydrometeorol.*, *16*(6), 2481–2500, doi:10.1175/jhm-d-15-0026.1.

Refsgaard, J. C. et al. (2014), A framework for testing the ability of models to project climate change and its impacts, *Clim. Change*, *122*(1–2), 271–282, doi:10.1007/s10584-013-0990-2.

Seibert, J. (2003), Reliability of Model Predictions Outside Calibration Conditions, *Nord. Hydrol.*, *34*(5), 477–492.

Thirel, G., V. Andréassian, and C. Perrin (2015), De la nécessité de tester les modèles hydrologiques sous des conditions changeantes, *Hydrol. Sci. J.*, *60*(7–8), 1165–1173, doi:10.1080/02626667.2015.1050027.

Thober, S., R. Kumar, N. Wanders, A. Marx, M. Pan, O. Rakovec, L. Samaniego, J. Sheffield, E. F. Wood, and M. Zink (2018), Multi-model ensemble projections of European river floods and high flows at 1.5, 2, and 3 degrees global warming, *Environ. Res. Lett.*, *13*(1), doi:10.1088/1748-9326/aa9e35.

Vormoor, K., D. Lawrence, M. Heistermann, and A. Bronstert (2015), Climate change impacts on the seasonality and generation processes of floods – projections and uncertainties for catchments with mixed snowmelt/rainfall regimes, *Hydrol. Earth Syst. Sci*, *19*, 913–931, doi:10.5194/hess-19-913-2015.

Wobus, C., E. Gutmann, R. Jones, M. Rissing, N. Mizukami, M. Lorie, H. Mahoney, A. W. Wood, D. Mills, and J. Martinich (2017), Climate change impacts on flood risk and asset damages within mapped 100-year floodplains of the contiguous United States, *Nat. Hazards Earth Syst. Sci.*, *17*, 2199–2211, doi:10.5194/nhess-17-2199-2017.

---

## Author Response (AR2)

Dear Dr. Peleg,

Thank you very much for the thorough assessment of our manuscript and for your invitation to resubmit our manuscript to HESS. Following your and the reviewers' comments, we rewrote the introduction to highlight the aim and novel contribution of the study. The main aim is to 'evaluate the extent to which models calibrated according to standard model calibration metrics such as the widely-used Kling–Gupta efficiency are able to capture flood spatial coherence and triggering mechanisms.' We specify that 'we first evaluate how well the different models capture local flood events following the current paradigm, secondly we expand the evaluation by analyzing how well the models capture spatial flood dependence, and finally we evaluate how the models capture flood triggering mechanisms.' In addition, we added a discussion section that is separate from the results section and expands the previous discussion of the results by discussing potential ways of improving model performance by developing flood-tailored calibration metrics, proposing spatial calibration metrics, identifying model structures representing the most important flood producing mechanisms, and investigating input uncertainty. We think that the major changes made to the introduction, discussion, and conclusions section help to clarify the storyline.
Please find our detailed answers to the reviewers' comments in our point-by-point response below. We hope that you find the revised version of our manuscript suitable for publication in HESS.

On behalf of all co-authors,

Manuela Brunner

**Editor's decision**

Dear Authors,

I have now received the reports of two referees, one (that did not revise the original text) suggested major revisions, while the other suggested minor revisions (although her/his comments read to me as moderate revisions). Reading the revised manuscript and the comments made by the reviewers, I conclude that additional changes to the text are needed before it can be considered for publication in HESS.

The main issue that I see here, is that the motivation, objectives and hypotheses of the study are not composing a clear storyline. This was already pointed by some of the reviewers in the first round of revision. In this study, a single objective function (KGE) is used for the model calibration, and you demonstrate that 4 different models are failing to represent the observed floods using this single-criteria objective function. In the conclusions, you suggested using multi-criteria objective functions for the calibration of the models if the focus is on representing flood events. This conclusion is not new – many studies in the past used multi-criteria objective functions to calibrate hydrological models to simulate floods, likely with a better match than can be obtained with KGE. Why have you chosen to calibrate the models using an objective function that is known (or can be expected) to fail to simulate flood events to begin with? What multi-criteria objective function/strategy could be used to calibrate hydrological models to better represent flood events (what strategies were used in the past and how they can be improved)? Will a multi-criteria objective function improves the match to flood events, or does some of the models that are presented here will still fail in reproducing flood events due to their internal structure? I am missing answers/discussion to these type of questions.

In my view, **the introduction, discussion and conclusions sections will require considerable text edits** to make the story clearer and more appealing to the readers of HESS, maybe also with minor

changes to the structure of the text. I will be happy to reconsider the revised paper after major revisions.

**Reply:** *Thank you very much for your clear opinion on how the manuscript can/should be improved. We rewrote the introduction to highlight the aim and novelty of the paper. The main aim is to 'evaluate the extent to which models calibrated according to standard model calibration metrics such as the widely-used Kling–Gupta efficiency are able to capture flood spatial coherence and triggering mechanisms' and the novelty is that 'we expand the evaluation by analyzing how well the models capture spatial flood dependence and how the models capture flood triggering mechanisms.' We focused the study on the Kling-Gupta efficiency metric because it is 'widely used in flood simulation studies (e.g. Hirpa et al., 2018; Huang et al., 2018; Thober et al., 2018; Brunner and Sikorska-Senoner, 2019; Harrigan et al., 2020) and has been shown to result in more accurate flood peak representations than the other widely used individual metric $E_{NS}$ [Mizukami et al., 2019]'.*

*With this study, another key aim is to raise awareness that notwithstanding the popularity of the KGE calibration metric in recent years, it may not be the best choice if one is interested in floods. This outcome may be appreciated by a small portion of the field, but we do not see evidence that it is widely appreciated, either from the literature or in the authors' own interactions with other researchers. There has, on the whole, been much aspirational discussion of more widespread use of hydrologic signatures, but there is also a continuing practice of defaulting to the generic KGE or NSE for many studies. These notably include some of the studies by the authors themselves, which in part produced these parameter sets. We added a new discussion section that significantly expands the discussion on potential multi-criteria objective functions for flood events, spatial model calibration, and the role of model structure in model performance. We think that the rewritten introduction, discussion, and conclusions sections convey a clear and useful story to the HESS readership.*

**Reviewer 1**

General comments

This study compares the efficiency of three lumped and one distributed models to simulated flood magnitude, timing and spatial coherence. The objective function used for calibration is Kling-Gupta efficiency (KGE). The results show that models tend to underestimate flood magnitude and not always simulate well flood timing. The authors conclude that using KGE for calibration has limited reliability for flood hazard assessment.

In general, the topic fits scope of the journal and will be of interest for the readers. However, the manuscript in its current form (after the revision) will still benefit from a more thorough revision. The main critical points (in my opinion) are:

1) The formulation and justification of the novel scientific contribution is still not clear. The review of previous studies in the Introduction indicates that "…to achieve further improvements in flood peak simulations, a broader range of application-specific evaluation metrics is typically required." (l.29-30, l.23-24). I agree with such formulation of current research gaps, but it is not in line with the objective function tested in the manuscript. If one would be interested in flood magnitude, timing and spatial connectivity, why one should use KGE for calibration? How does it account for such specific evaluation metrics, i.e. flood seasonality or spatial coherence?

**Reply:** *Thank you for highlighting the need to better work out the novelty of our study and to justify the use of $E_{KG}$ for model calibration. The main aim is to 'evaluate the extent to which models calibrated according to standard model calibration metrics such as the widely-used Kling-Gupta efficiency are able to capture flood spatial coherence and triggering mechanisms' while the novelty is that 'we expand the evaluation by analyzing how well the models capture spatial flood dependence and how the models capture flood triggering mechanisms.' We focused the study on the Kling-Gupta efficiency metric because it is 'widely used in flood simulation studies (e.g. Hirpa et al., 2018; Huang*

*et al., 2018; Thober et al., 2018; Brunner and Sikorska-Senoner, 2019; Harrigan et al., 2020) and has been shown to result in more accurate flood peak representations than the other widely used metric $E_{NS}$ [Mizukami et al., 2019]'. We chose $E_{KG}$ for calibration to highlight that the widely used metric might not lead to accurate flood simulations as often assumed. The point is exactly that $E_{KG}$ is often used as a metric in flood simulation studies despite the fact that it may lead to suboptimal model performance with respect to floods. With this study, we want to raise awareness for exactly this issue. By raising awareness of this matter, we hope to inspire other researchers to propose alternative calibration metrics. In the newly created discussion section, we present ideas on how flood simulations could be improved by developing alternative calibration metrics, developing spatial calibration metrics, identifying suitable model structures, and improving precipitation input data.*
**Modification: p1. l.5-8, p.2 l.27-34 and l.49-54**

2) The title is misleading. The main message of the paper, in its current form, is about the value of KGE for calibration of hydrologic models (if flood impact assessment is the main purpose). There is no assessment how the models describe and simulate different flood generation processes and which factors control their performance. So based on presented results it is difficult to interpret to what extent and how are the selected models suitable for flood impact assessment. The results are more about the accuracy of selected way (i.e. using lumped models, KGE for calibration, etc.).
**Reply:** *Thank you for pointing out the need for revising the current title to better reflect the main message of the study. The revised version goes beyond discussing the value of $E_{KG}$ as a calibration metric by also discussing the role of model structure, which is enabled by the comparison of the performance of four different models. We chose the following new title: 'Flood spatial coherence, triggers and performance in hydrological simulations: large-sample evaluation of four streamflow-calibrated models', which highlights that the paper is about model evaluation for floods and reflects the focus on spatial flood characteristics and the representation of flood drivers.*
**Modification: title**

3) The significance of the results is not clear. I'm not sure if for practical applications, a lumped model will be used or should be recommended. Perhaps a consistent assessment/evaluation of the difference between lumped and distributed type of models will be interesting (e.g. for HBV and mHM).
**Reply:** *We agree that a more in-depth discussion of the results was needed in order to highlight their significance. We therefore separated the Results section from a newly created Discussion section. In this new section, we discuss the findings and propose potential ways of improving flood simulations by moving away from standard calibration metrics such as $E_{KG}$, by identifying suitable model structures, and by improving the quality of input precipitation. We agree that an explicit assessment of the role of model type (lumped vs. distributed) would be interesting and we think that such an assessment is out of scope of this paper.*
**Modification: p.12 l.232-p.17 l.329**

4) The design of the experiment reads more as a collection of available analyses and not results from initially clearly defined research question/hypothesis. I agree with previous reviews that using different time periods for calibration and using different model input datasets can have some impact on the results and the interpretation of results (including individual catchments) will be more consistent if the same data and time periods will be used. The authors claim that both datasets describe the observed climate, but are they identical also for individual extreme events?
**Reply:** *We reworked the introduction to highlight the main aim of the study, i.e. 'This study evaluates the extent to which models calibrated according to standard model calibration metrics such as the widely-used Kling--Gupta efficiency are able to capture flood spatial coherence and triggering mechanisms', and stress that 'we expand the evaluation by analyzing how well the models capture spatial flood dependence and how the models capture flood triggering mechanisms.' To achieve this goal, we use simulations generated in previous studies, which serve as a proxy for simulations that*

*would be typically used in research studies on flooding because they used best possible calibration settings given past computer and resources availability. We agree that ideally the same precipitation input would have been used for all models, which was unfortunately not possible because the simulations our study is based on were derived by different authors. The two datasets, however, were derived from observed precipitation and temperature, and have been shown to result in similar mean daily precipitation fields [Newman et al., 2015]. We therefore consider them similar enough to allow for a direct comparison of the model outputs resulting from the different input datasets.*
**Modification: p.2 l.49-54**

5) The methodology is not rigorously described. It will be very difficult (if even possible) to reproduce/repeat the presented analysis (based on given information). Numerous information is missing, e.g., how the initial values were set, what were the ranges of calibrated model parameters and parameters of automatic calibration algorithm. It will be interesting to present, e.g. in appendix, the final model parameters and efficiencies for individual catchments. This will allow to assess the interpretation made.

**Reply:** *Our study seeks to extract information through the pooling of different modeling results. For this, we used streamflow simulations derived in previous studies by Melsen et al. (2018) and Mizukami et al. (2019) as described in the Methods section. Thus, we did not have complete control over the design of the experiment. As specified in the data availability section, model simulations can be obtained by the main authors of these previous studies. Details on the model calibration procedures used in these studies can be found in the references. Melsen et al. (2018) provide information on the parameter boundaries used in the Sobol-based Latin hypercube sampling in Tables C1-C3. Mizukami et al. (2019) did not provide specific parameter ranges in the original paper published. We think that providing model parameters for 671 catchments is infeasible even in the appendix as this would produce a large amount of additional pages, which in our opinion hardly anyone would look at.*

6) I think that comparing lumped with distributed models can bring some more interesting results than are presented in its current form. What is the impact of lumping on the results? Are the differences in model efficiency related to the size of the basin? I would expect that using lumped models in larger catchments cannot describe well floods from convective rainfalls.

**Reply:** *Thank you for suggesting this additional analysis. Figure 1 shown in this review shows flood model errors at individual sites for five different catchment size classes. Model performance does not seem to depend on catchment size and we can not identify significant differences in outcomes between distributed models (mHM) and lumped models (other three models). The fact that we can draw similar conclusions from both lumped and distributed models strengthens the argument being made. The lumped/distributed impact questions would be a good topic for a follow on study, though there is some literature on that topic already (e.g the Distributed Model Intercomparison Project DMIP study: https://www.weather.gov/owp/oh_hrl_distmodel_dmip_draft).*

[Figure]

*Figure 1: Model errors per catchment size class computed over the period 1981-2008: <100 km²
(green, 93 catchments), >=100 and <250 km² (yellow, 115), >=250 and <500 km² (blue, 112), >=500
and <1000 km² (pink, 93), and >1000 km² (violet, 75). Errors are shown for (i) number of events (error
in number of events), (ii) magnitude (mean relative error in %), and (iii) timing (mean absolute error in
days) for the four models (1) SAC, (2) HBV, (3) VIC, and (4) mHM. The boxplots are composed of one
value per catchment belonging to the respective catchment size class.*

7) As a reader, I would be likely more interested in seeing where (in which catchments and why?) the
models work well, rather than to conclude that in general they underestimate magnitude or do not
represent well the timing or spatial patterns. So presenting some deeper analysis of the factors
controlling the performance will be helpful and interesting.

**Reply:** *We agree that providing model evaluations for different types of catchments is interesting. We
therefore perform model evaluations for 5 types of streamflow regimes (see Figure 1) as shown in
Figures 2, 3, 4, and 7. This regime-specific analysis allows us to conclude that 'model performance is
generally worst in catchments with intermittent regimes while it is highest for catchments with a
strong seasonality such as a melt and New Year's regime.' We highlight the regime-specific analysis in
the Methods section by saying: 'To provide insights with respect to where model performance is
better/worse, we provide model evaluation results for five different streamflow regime types, which
have been shown to be distinct in their flood behavior: 1) Intermittent, 2) weak winter, 3) strong
winter, 4) New Year's, and 5) melt. Catchments with intermittent regimes experience floods mainly in*

*spring and summer, those with weak winter regimes in winter and spring, those with strong winter regimes in winter, those with a New Year's regime around New Year, and those with a melt-dominated regime in spring because of snowmelt.'*
**Modification: p.4 l.95-100**

Specific comments

1) Abstract, l.13: "…models calibrated on integrated metrics such as …have limited reliability…". This is a general conclusion which is not supported well with the presented results. I would suggest to remove "such as". I think if one combines flood magnitude, seasonality and spatial coherence into an integrated metrics (objective function) for calibration, the results can be different.
**Reply:** *Thank you for the rephrasing suggestion, which we adopted.*
**Modification: p.1 l.16**

2) Data. I like the assessment based on large dataset and subsequent split/grouping of results into some relevant groups of catchments. It is however not clear how are flood generation processes (e.g. flood types) linked with selected groups of regimes? If the objective is about the suitability of models to represent floods (magnitude, seasonality, …) it will be interesting to see results for different flood generation mechanisms, i.e. how or if the models differ in simulating snowmelt floods, or floods from convective rains, etc.
**Reply:** *Thank you for pointing out to better introduce the regime-specific analysis. We specify in the Methods section that 'To provide insights with respect to where model performance is better/worse, we provide model evaluation results for five different streamflow regime types, which have been shown to be distinct in their flood behavior: 1) Intermittent, 2) weak winter, 3) strong winter, 4) New Year's, and 5) melt (Figure 1; Brunner et al.,* 2020*). Catchments with intermittent regimes experience floods mainly in spring and summer, those with weak winter regimes in winter and spring, those with strong winter regimes in winter, those with a New Year's regime around New Year, and those with a melt-dominated regime in spring related to snowmelt.' We agree that further distinguishing between different flood generation types would be very interesting as well. However, we think that such an assessment would be a separate (classification/clustering) study in itself.*
**Modification: p.4, l.95-100**

3) Forcing. Which version of Daymet is used? Why not to use only one dataset for all the models?
**Reply:** *We based this study on previously published work, which used best possible calibration settings given past computer and resources availability, and scope. These studies used slightly different model inputs as described in the Methods section. The two datasets, however, were derived from observed precipitation and temperature, and have been shown to result in similar mean daily precipitation fields [Newman et al., 2015]. Melsen et al. (2018) used Daymet version 2.1. for their simulations with SAC, HBV, and VIC. Recognizing some inconsistencies in the different sources of data, we nonetheless felt that extending our analysis to cover the multiple models would make our findings more robust.*

4) The term "event": By using term flood event, do you mean day of the flood peak? The same for precipitation. Is the event precipitation representing mean daily precipitation for the day of the peak? Some flood events (e.g. from snowmelt) can last several days. How sensitive/representative are the characteristics extracted only for the day of the peak?
**Reply:** *Thank you for highlighting the need for clarifying the meaning of the term 'event' and for specifying how corresponding rainfall, snowmelt, and soil moisture were identified. We specify that by a peak-over-threshold flood event, we mean peak discharge, and that precipitation, snowmelt, and soil moisture were identified for the day of peak discharge. To identify how sensitive the results shown in Figure 5 are to the aggregation level (i.e. 1 day), we performed the same analysis also with 3-day precipitation and snowmelt sums. The results look almost identical to the ones presented for*

*the 1-day aggregation level.*
**Modification: p.6 l.119, p.7 l.160**

5) Beta model parameter (l.170). It will be interesting to present model parameters for individual catchments, because otherwise the interpretation made reads more as speculation (it is not justified by presented results).
**Reply:** *Thank you for indicating the need to distinguish between results and their discussion. To do so, we created a new Discussion section. We moved the statement about the role of the beta parameter to this new discussion section to highlight that we use it to interpret the results presented in Figures 3 and 4.*
**Modification: New discussion section (p.12-17)**

6) L.218-219. In my opinion HBV model can describe the surface runoff. Conceptual it is represented by the outflow from the upper reservoir (describing by k0 model parameter).
**Reply:** *We think that this statement is correct because [Bergström, 1976] wrote: 'All versions of the HBV-model are lacking components for direct surface runoff, as the water is controlled by the conditions in the soil moisture zone before any runoff can be generated.'*

**Reviewer 2**

The authors responded to most of my concerns, but some issues are left.
**Reply:** *Thank you very much for taking the time to write this second review.*

[1] The title is still problematic. As far as I can tell, there is no "flood impact assessment" performed in this study. Why is it in the title? The study assess flood flows, so why is this not the title of the paper? Flood impact assessment would require a direct connection to the actual implication of flooding, such as flood inundation, damage to houses etc. These aspects are not part of the study, so why is the title focusing on this issue?
And, if the focus is on assessing the value of KGE as calibration metric, then why is this not in the title? The title "Evaluating the suitability of hydrological models for flood impact assessments", is still much broader than what this very focused study actually does.
**Reply:** *Thank for pointing out the need to further improve the title. We agree that we are not performing a flood impact assessment and that talking about flood simulations instead would be more appropriate. We propose the following revised title: 'Flood spatial coherence, triggers and performance in hydrological simulations: large-sample evaluation of four streamflow-calibrated models', which stresses that this study is about model evaluation and that the focus is on both local and spatial flood characteristics.*
**Modification: Title**

[2] (line 140) The use of split sample schemes should include a reference back to Klemes (1986, HSJ, https://www.tandfonline.com/doi/abs/10.1080/02626668609491024 ) who introduced the idea.
**Reply:** *Thank you for indicating the need to cite the original reference to the split sample testing idea. We added the reference to the text.*
**Modification: p.2 l.48, p.7 l.154**

[4] Section 3.1: I asked previously why HBV results are so poor and I am still confused by it. It would be useful for the discussion section of this paper to more closely compare the results obtained here to previous studies across the USA. For example, Kollat et al. (2012, WRR, https://agupubs.onlinelibrary.wiley.com/doi/full/10.1029/2011WR011534) calibrated the HBV model across all MOPEX catchments and found Nash Sutcliffe Efficiency values much higher than what would be expected based on the results of the current study (see their Figure 9A). Why the discrepancy? Kollat et al. (2012) performed extensive MO-calibration whereas the current study used

a LHS sampling strategy. So, is part of the result of the current study is due the chosen calibration approach?

**Reply:** *Thank you for indicating this reference. We looked the $E_{NS}$ values presented in Figure 9A of Kollat et al. (2012). This figure shows that roughly 50% of the catchments show a $E_{NS}$ value >0.75. If we determine the percentage of catchments in our dataset that has $E_{KG}$ (we did not use $E_{NS}$ to identify best parameter sets) values >0.75, we get 47%. Similarly, roughly 90% of the catchments in the Kollet et al. (2012) study show $E_{NS}$ values above 0.5, the same percentage of catchment that also exceeds $E_{KG}$ values of 0.5 in our study. We therefore argue that the model performance of the HBV model used in our study is as good as the performance of the model calibrated in the Kollat et al. (2012) study.*

[5] Other studies have disaggregated KGE to understand what controls the bias in the KGE terms. E.g. Gudmundsson et al. (2012, WRR, https://agupubs.onlinelibrary.wiley.com/doi/full/10.1029/2011WR010911) found that one significant control on water balance error seemed to be precipitation data error – model predictions in catchments with significant elevation difference (where at least across Europe, precipitation measurements are expect to be less good) were performing poorer. Did the authors find similar patterns? Would elevation difference be a good way to see whether rainfall is indeed a likely problem for the study catchments in this paper?

**Reply:** *As suggested, we checked whether model performance in terms of $E_{KG}$ is related to elevation (Figure 2 shown in this response to the reviewers). High-elevation catchments have generally better performance than low elevation catchments, but present a confounding factor in that higher catchments more often experience snow, which has generally a positive influence on model calibration. This finding suggests that precipitation errors, which are typically higher in high-elevation catchments due to less dense measurement networks, are not the main determinant of model performance.*

[Figure]

*Figure 2: $E_{KG}$ vs. elevation for the four models tested: SAC, HBV, VIC, and mHM.*

[3] I am still confused by the authors conclusion that "Our model comparison shows that all flood characteristics are not equally well represented by models calibrated with the widely used Kling–Gupta efficiency metric." – OK, but very likely this is true for any metric given the extensive experience with multi-objective model calibration in hydrology, where a regular finding is that any single metric produces a focused result. So, what multi-objective strategy do we need to improve this

problem? And what relevant trade-offs exists (e.g. Kollat et al., 2012)? The authors suggest that multi-objective calibration is the way forward, in which case a better review of this very rich multi-objective literature in hydrology would be nice (given that this topic has been explored for over 20 years). Currently there are only a couple recent references, which do not do the topic justice – even if narrowed down to those studies focusing on calibration for flood prediction.

**Reply:** *Thank you for highlighting the need to expand the discussion on multi-objective calibration strategies. We significantly expanded the discussion of multi-objective calibration in the context of flood modeling and also added references to the more classical literature on multi-objective calibration not necessarily targeted at floods.*

**Modification: p.16 l.282-295**

The next sentence suggests a much wider conclusion: "The number of floods, flood magnitude, and timing are not always well captured by hydrological models in many catchments." It would be good if the authors were to formulate their conclusions more carefully. Given that the authors have a very narrow focus in this study (which is fine) – to show that calibrating to KGE does not lead to a good reproduction of all flood characteristics – it would be good to formulate their conclusions with a similar focus to avoid that others misuse their conclusions.

**Reply:** *Thank you for pointing out the need to more concisely phrase the conclusions. We changed this sentence to: '
[revised manuscript text omitted]

---

## Author Response (AR3)

Dear Dr. Peleg,

Thank you very much for your assessment. We considered the comments made by Referee #4 by implementing the requested edits and moved the material presented in the appendix to the Supplementary Material (i.e. a separate new PDF). Thank you very much for reconsidering our article for publication in HESS.

On behalf of all co-authors,

Manuela Brunner

**Editor's decision**

Please consider the comments made by Anonymous Referee #4. In addition, please consider presenting the Appendix as Supplementary Material. I am looking forward to reading the revised paper.
**Reply:** *We created a new PDF, which contains the Supplementary Material previously provided in the Appendix.*

**Reviewer 4**

I want to thank you for considering most of my comments and revisions made. The story has been significantly improved, including formulations of the aim and scientific contribution of the study. Only one general comment (point 5 in the previous review) has not been addressed adequately, but I think it is essential for supporting interpretations made. The results (and the response) indicate no clear pattern of the difference between distributed and lumped models. In particular, the comparison and difference between HBV and MHM do not show any relation to the size of the basin. But the different results of these two models do not only represent the difference between the lumped vs distributed format, but also in the datasets used for driving the models. The manuscript indicates (and the authors believe) that both datasets adequately describe the observed climate. Still, there is no support/evidence that these datasets also provide identical/similar patterns of event precipitation characteristics (e.g. magnitude, antecedent sum, duration, etc.). The differences in the precipitation differences can mask the differences between lumped and distributed models. A similar comment also applies for the difference in the calibration periods, but this has a likely smaller effect.
**Reply:** *Thank you very much for acknowledging the improvements of our manuscript. We agree that the slight differences in precipitation products may mask potential differences between lumped and distributed models. However, although mHM is distributed, the meteorological forcing data are spatially averaged over a basin and all basins are relatively small (median size of ~300km², mean of ~500km²), i.e. cover a handful of model grid cells, meaning that model distributiveness would have little impact on simulated flows. Still, we agree that for large basins with 1000s of grid cells, results derived for lumped and distributed versions might look more different than in the present study. In any case, the potential masking effect of differences in precipitation products does not impact the conclusions drawn in our study because we do not aim at highlighting differences between lumped and distributed models. Rather, our goal was to pool different models (both lumped and distributed) to create a large sample size to assess how models calibrated on KGE represent spatial flood characteristics and flood triggers. We added the following acknowledge of the advantages and disadvantages of model pooling to the methods section: 'Basing the study on four different modeling efforts has the advantage of enlarging the sample size on which conclusions can be drawn, but the disadvantage that the models were not run as part of a controlled study, with consistent forcings, calibration periods, and parameter selection.' In addition, we highlight the potential importance of input uncertainty on model results in the manuscript by stating that: 'In addition, the uncertainty of*

*the precipitation product used to drive a hydrological model can lead to differences in observed and simulated flows (teLinde et al. 2007, Renard et al. 2011). Precipitation products may show observation uncertainties (McMillan et al. 2012) and underestimate extreme rainfall or the spatial dependence of extreme precipitation at different locations because spatial smoothing or averaging during the gridding process reduces variability (Haylock et al. 2008, Risser et al. 2019).'*
**Modification: p4, l.78-80 and p17, l.325-329**

**Specific comments**

Abstract: please remove repletion of "the widely-used Kling–Gupta efficiency" (l.6-7, l.9-10)
**Reply:** *We removed 'widely-used' at one of these instances.*

Abstract: repetition l.13, l.14; "not necessarily…"
**Reply:** *We replaced the second 'not necessarily' by 'not always'.*
**Modification: p1, l.13**

Discussion, l.235 "The results presented in this study demonstrate that simulating floods using hydrological models …". Please be more specific…I think, the results show only the case if the models are calibrated to KGE. I think if you use seasonality or flood magnitude in the objective function, the results can be different.
**Reply:** *Thank you for pointing out the need to be more specific here. We specify that: '
[revised manuscript text omitted]